# Efficient Meta Learning via Minibatch Proximal Update

**Pan Zhou**[*]  **Xiao-Tong Yuan**[†]  **Huan Xu**[‡]  **Shuicheng Yan**[△]  **Jiashi Feng**[*]

[*] Learning & Vision Lab, National University of Singapore, Singapore
[†] B-DAT Lab, Nanjing University of Information Science & Technology, Nanjing, China
[‡] Alibaba and Georgia Institute of Technology, USA
[△] YITU Technology, Shanghai, China
pzhou@u.nus.edu  xtyuan@nuist.edu.cn  Huan.xu@alibaba-inc.com  {eleyans, elefjia}@nus.edu.sg

## Abstract

We address the problem of meta-learning which learns a prior over hypothesis from a sample of meta-training tasks for fast adaptation on meta-testing tasks. A particularly simple yet successful paradigm for this research is model-agnostic meta-learning (MAML). Implementation and analysis of MAML, however, can be tricky; first-order approximation is usually adopted to avoid directly computing Hessian matrix but as a result the convergence and generalization guarantees remain largely mysterious for MAML. To remedy this deficiency, in this paper we propose a minibatch proximal update based meta-learning approach for learning to efficient hypothesis transfer. The principle is to learn a prior hypothesis shared across tasks such that the minibatch risk minimization biased regularized by this prior can quickly converge to the optimal hypothesis in each training task. The prior hypothesis training model can be efficiently optimized via SGD with provable convergence guarantees for both convex and non-convex problems. Moreover, we theoretically justify the benefit of the learnt prior hypothesis for fast adaptation to new few-shot learning tasks via minibatch proximal update. Experimental results on several few-shot regression and classification tasks demonstrate the advantages of our method over state-of-the-arts.

## 1   Introduction

Meta-learning [1, 2, 3], a.k.a. learning-to-learn [4], is an effective approach for learning fast from small amount of data, with many successful applications witnessed to regression/classification [5, 6, 7, 8, 9, 10] and reinforcement learning [6, 11, 12, 13]. It assumes access to a distribution of tasks, each of which could be a learning problem (e.g. classification), and then learns from a finite set of sample meta-tasks. Specifically, meta-learning contains a meta-learner which is a trainable learning hypothesis or algorithm to extract knowledge from all observed meta-tasks and facilitate the learning a learner for a potentially unseen meta-task with only a few samples. The current meta-learners can be grouped into metric-based approaches [14, 8, 15, 16] which learn the similarity metrics among samples, memory-based methods [17, 7, 18] which use memory models, e.g. neural Turing machines [19] and long short-term memory [20], to store important training samples or learn a fast adaptation algorithm, and optimization-based approaches [6, 5, 9, 10, 21] that learn a good parameter initialization or regularization for fast adaptation to new tasks. Compared with metric-based approaches which are more suitable for non-parametric learners, and memory-based methods which are designed case-by-case, optimization based approaches are simpler but also more general, and thus have been applied to various applications without lots of tailors [6, 9].

MAML [6] is a recent leading approach of optimization-based meta-learning. In principle, MAML aims to estimate a good parameter initialization $w$ of a network such that for a randomly sampled task

$T$ with corresponding loss $\mathcal{L}_{D_T}(\boldsymbol{w})$, the meta-loss $\mathcal{L}_{D_T}(\boldsymbol{w} - \eta\nabla\mathcal{L}_{D_T}(\boldsymbol{w}))$ is small. This method is compatible with any model trained with gradient descent, a.k.a. model-agnostic, and has been showed to be effective in many classification and reinforcement learning applications. However, for gradient based meta-optimization, MAML requires computing second-order derivative introduced by the intra-task gradient descent step $\boldsymbol{w} - \eta\nabla\mathcal{L}_{D_T}(\boldsymbol{w})$ which is computationally inhibitable for large networks. To resolve this issue, first-order approximates of MAML [6, 10] have been developed to avoid the estimation of second-order derivative. For example, first-order MAML (FOMAML) [6] directly ignores the second-order derivative terms in MAML and Reptile [10] approximates the gradient in MAML by the gradient sum of several gradient descent steps on a joint training model. Though exhibiting impressive scalability and accuracy in some applications, the convergence and generalization behavior of these variants is under explored and remains largely mysterious, especially for non-convex problems. Indeed for few-shot classification, as shown in Sec. 4 that the first-order approximate approaches have been witnessed to suffer from generalization performance degradation, e.g. on the Omniglot [10] and tieredImageNet datasets, due to gradient approximation steps.

To remedy the deficiency of MAML, we consider the minibatch proximal update regularized by prior hypothesis parameters $\boldsymbol{w}$, i.e., $\min_{\boldsymbol{w}_T} \mathcal{L}_{D_T}(\boldsymbol{w}_T) + \frac{\lambda}{2}\|\boldsymbol{w}_T - \boldsymbol{w}\|_2^2$, which aims to learn the optimum hypothesis parameter $\boldsymbol{w}_T^*$ of task $T$ around a prior hypothesis $\boldsymbol{w}$. Such a mechanism can leverage the *good* prior $\boldsymbol{w}$ to facilitate the learning of task $T$ with only a few samples, as it tells the learner where the optimum hypothesis parameter $\boldsymbol{w}_T^*$ roughly locates in the solution space. Then, how to efficiently learn the prior hypothesis parameters $\boldsymbol{w}$ becomes a crucial problem. Through the lens of online convex optimization, a follow-the-meta-regularized-leader method has been proposed for estimating such a prior hypothesis via online meta-learning [21]. In a concurrent work [22], for linear prediction models, a similar idea of minibatch proximal update has been explored inside a framework of online convex meta-learning. Differently and independently, we develop an SGD based meta-optimization algorithm for efficient meta-learning via minibatch proximal update, with provable guarantees established for a broader range of convex and non-convex learning problems than those considered in [21, 22].

**Our contributions.** In this paper, we present Meta-MinibatchProx as a generic stochastic meta-learning approach for learning a good prior hypothesis for minibatch proximal update. The idea is to view the prior hypothesis as an unknown meta-parameter, and learn it via minimizing the empirical risk over a set of minibatch proximal update based meta-training tasks. More specifically, in the off-line setting, for both convex and non-convex loss functions we seek to minimize the meta-empirical-risk $L_n(\boldsymbol{w}) = \frac{1}{n}\sum_{i=1}^n \phi_{D_{T_i}}(\boldsymbol{w})$ over a set of meta-training tasks $\{T_i\}$ sampled from a task distribution $\mathcal{T}$, where $\phi_{D_{T_i}}(\boldsymbol{w}) := \min_{\boldsymbol{w}_{T_i}} \left\{ \mathcal{L}_{D_{T_i}}(\boldsymbol{w}_{T_i}) + \frac{\lambda}{2}\|\boldsymbol{w}_{T_i} - \boldsymbol{w}\|_2^2 \right\}$ is the meta-loss of task $T_i$ determined by the minibatch proximal update. While in the online setting, we alternatively seek to minimize the expected risk function $L(\boldsymbol{w}) = \mathbb{E}_{T_i \sim \mathcal{T}}[\phi_{D_{T_i}}(\boldsymbol{w})]$.

A key observation of our approach is that the gradient of the meta-loss evaluated at each meta-training task $T_i$ can be expressed in closed-form as $\nabla\phi_{D_{T_i}}(\boldsymbol{w}) = \lambda(\boldsymbol{w} - \boldsymbol{w}_{T_i}^*)$, where $\boldsymbol{w}_{T_i}^*$ is an optimal hypothesis output by the minibatch proximal update. This reveals that the gradient evaluation of meta-loss can be computed via solving the intra-task minibatch proximal update problem without necessarily accessing the Hessian information of the empirical risk. This in turn paves the way for employing any off-the-shelf first-order method for meta-optimization. In our implementation, we simply use stochastic gradient descent (SGD) with provable convergence guarantees established simultaneously for convex and non-convex problems.

Moreover, we theoretically show that the quality of the prior hypothesis regularizer plays an important role in controlling the excess risk (a.k.a. population sub-optimality) of minibatch proximal update in the testing phase. More specifically, given a learned hypothesis $\boldsymbol{w}$, the expected excess risk of convex minibatch proximal update on a training sample set of size $K$ is upper bounded at the order of $\mathcal{O}\left(\frac{1}{\sqrt{K}}\sqrt{\mathbb{E}_{T\sim\mathcal{T}}\left[\|\boldsymbol{w} - \boldsymbol{w}_{T,E}^*\|_2^2\right]}\right)$, where $\boldsymbol{w}_{T,E}^*$ represents the population optimal hypothesis of any given meta-task $T \sim \mathcal{T}$. This guarantees that if the hypothesis $\boldsymbol{w}$ is close to each task-specific optimal hypothesis in average, then adapting $\boldsymbol{w}$ as a prior hypothesis regularizer in minibatch proximal update to a new task with only a few samples will have better generalization ability than using random initialization for adaptation. This further justifies the motivation of learning to prior hypothesis transfer for efficient minibatch proximal learning. Extensive experimental results well demonstrate the advantages of our approach in few-shot deep learning problems.

## 2 Related Work

**Optimization based meta-learning.** The family of optimization based meta-learning approaches aims to directly learn a good parameter initialization or regularization for future optimization and has gained a lot of attention recently thanks to its simplicity, versatility and effectiveness [6, 5, 9, 10, 21]. As a representative method in this line, MAML [6] tries to estimate an initialization network parameter such that for a randomly sampled new task the network can be fine-tuned in one or few steps of minibatch gradient descent. To avoid the computation of second-order derivatives, first-order approximations of MAML have been developed including FOMAML [6] and Reptile [10]. For lifelong learning, an follow-the-meta-leader extension of MAML has been studied in the setting of online learning [23]. Alternative to meta-initialization, the meta-regularization approaches have gained recent interest mostly due to their provable strong guarantees on statistical learning and computation efficiency [24, 25, 26, 27, 21]. The most closely relevant work to ours are [22, 27] and [21] which also consider the prior hypothesis biased regularized empirical risk minimization for intra-task learning. Different from [22, 27] which focus on linear regression/classification with convex loss functions, our approach is developed inside a broader context of convex and non-convex statistical learning and thus is of more practical interest especially in deep learning. In contrast to the online convex meta-learning framework developed in [21], we use a simple yet scalable paradigm of minimatch-prox within SGD for stochastic meta-optimization which is particularly friendly for computational and statistical complexity analysis in both convex and non-convex settings.

**Minibatch proximal, hypothesis transfer, and multi-task learning.** As a building block of our approach, the minibatch proximal update method has been studied in different contexts including online passive-aggressive learning [28], asynchronous stochastic gradient optimization [29], and communication-efficient distributed learning [30], to name a few. Minibatch proximal learning is also identical in principle to biased regularized hypothesis transfer learning, which has been explored experimentally with success in many applications [31, 32, 33] and theoretically with rigourous guarantees [34, 35]. In the context of multi-task learning, a biased regularized approach was considered in [36] to learn many related tasks simultaneously such that the learnt hypotheses should be close to their mean. In contrast, inspired by the strong power of meta-learning for learning how to learn, we seek to learn a good prior hypothesis as proximity regularizer for future task learning.

## 3 Meta-Learning via Minibatch Proximal Update

In this section, we introduce the Meta-MinibatchProx method along with its optimization algorithm. We also provide a analysis to justify the benefit of the learned prior hypothesis regularizer.

### 3.1 Meta-Problem Formulation

Given sample space $\mathcal{X}$ and target space $\mathcal{Y}$, our primal goal is to learn good prior hypothesis parameters $\boldsymbol{w}$ for a class of parameterized hypothesis $f : \mathcal{X} \mapsto \mathcal{Y}$ such that when facing with a new task $T$, the task-specific hypothesis parameters $\boldsymbol{w}_T$ can be quickly learned from a minibatch $D_T = \{(\boldsymbol{x}_1, \boldsymbol{y}_1), \cdots, (\boldsymbol{x}_K, \boldsymbol{y}_K)\}$ of $K$ random samples via the following minibatch proximal update:

$$\min_{\boldsymbol{w}_T} \mathcal{L}_{D_T}(\boldsymbol{w}_T) + \frac{\lambda}{2} \|\boldsymbol{w}_T - \boldsymbol{w}\|_2^2, \tag{1}$$

where $\mathcal{L}_{D_T}(\boldsymbol{w}_T) = \frac{1}{K} \sum_{(\boldsymbol{x}, \boldsymbol{y}) \in D_T} \ell(f(\boldsymbol{w}_T, \boldsymbol{x}), \boldsymbol{y})$ is the empirical risk for task $T$ and $\lambda$ is a regularization constant. The loss function $\ell(f(\boldsymbol{w}_T, \boldsymbol{x}), \boldsymbol{y})$ measures the discrepancy between the prediction $f(\boldsymbol{w}_T, \boldsymbol{x})$ and the ground truth $\boldsymbol{y}$, e.g. the mean-square-error in regression and the cross-entropy loss in classification. To learn a prior hypothesis for minibatch proximal update, given a meta task distribution $\mathcal{T}$, it is natural for us to consider the online (stochastic) meta-learning problem:

$$\min_{\boldsymbol{w}} \mathbb{E}_{T \sim \mathcal{T}} \Big[ \min_{\boldsymbol{w}_T} \Big\{ \mathcal{L}_{D_T}(\boldsymbol{w}_T) + \frac{\lambda}{2} \|\boldsymbol{w}_T - \boldsymbol{w}\|_2^2 \Big\} \Big]. \tag{2}$$

Problem (2) contains two levels of learning: for the inner level of intra-task learning, it aims to find the task-specific optimal hypothesis parameters $\boldsymbol{w}_T$ of task $T$ around the prior hypothesis $\boldsymbol{w}$, while for the outer level of inter-task learning, the model leverages the biased optimal hypothesis $\boldsymbol{w}_T$ to tune $\boldsymbol{w}$ such that $\boldsymbol{w}$ has small distance in average to all $\boldsymbol{w}_T$. By optimizing the inner and outer problems sufficiently, the estimated regularizer $\boldsymbol{w}$ can be expected to be close to the optimal

---

**Algorithm 1** SGD for Meta-MinibatchProx

---

**Input:** Initial point $\boldsymbol{w}_0$, learning rate $\{\eta_s\}$.
**for** $s = 0$ **to** $S - 1$ **do**
    Uniformly randomly select a mini-batch of task set $\{T_i\}$ of size $b_s$ from the observed $n$ tasks.
    **for** $T_i \in \{T_i\}$ **do**
        Compute an $\epsilon_s$-approximate stable minimizer $\boldsymbol{w}_{T_i}^s$ to the within-meta-task problem
        $\min_{\boldsymbol{w}_{T_i}} g(\boldsymbol{w}_{T_i}) := \mathcal{L}_{D_{T_i}}(\boldsymbol{w}_{T_i}) + \frac{\lambda}{2}\|\boldsymbol{w}_{T_i} - \boldsymbol{w}^s\|_2^2$ such that $\|\nabla g(\boldsymbol{w}_{T_i}^s)\|_2^2 \leq \epsilon_s$.
    **end for**
    Update the meta parameter $\boldsymbol{w}^{s+1} = \boldsymbol{w}^s - \eta_s \lambda (\boldsymbol{w}^s - \frac{1}{b_s}\sum_{i=1}^{b_s} \boldsymbol{w}_{T_i}^s)$.
**end for**
**Output:** the parameter initialization $\boldsymbol{w}^S$ of model $f$.

---

hypothesis of task $T$ sampled from $\mathcal{T}$ and thus can serve as a good prior hypothesis for a new minibatch proximal update task. Usually we are only provided with $n$ observed tasks $\{T_i\}_{i=1}^n$ drawn from $\mathcal{T}$, and thus seek to minimize the following off-line (empirical) version of problem (2) which we call Meta-MinibatchProx:

$$\min_{\boldsymbol{w}} F(\boldsymbol{w}) := \frac{1}{n}\sum_{i=1}^n \min_{\boldsymbol{w}_{T_i}} \left\{ \mathcal{L}_{D_{T_i}}(\boldsymbol{w}_{T_i}) + \frac{\lambda}{2}\|\boldsymbol{w}_{T_i} - \boldsymbol{w}\|_2^2 \right\}. \tag{3}$$

In Sec. 3.2 and Sec. 3.3, we will focuses on the above off-line setting for algorithm design and analysis, since in most applications (e.g. image classification) the number of training tasks is usually finite though the number $n$ may be large. We emphasize that all the convergence and generalization guarantees established in the off-line setting can be easily extended to the online stochastic setting, as is outlined in details in Appendix A.3.

To compare with MAML, Meta-MinibatchProx is also model-agnostic because it is compatible with a broad range of statistical learning models. On top of that, our model is algorithm-agnostic in the sense that the intra-task subproblem can be optimized using virtually any off-the-shelf machine learning optimization algorithms. This makes Meta-MinibatchProx more flexible than MAML which by design relies on minibatch stochastic gradient descent to fine-tune the meta-initialization $\boldsymbol{w}$.

Note that MAML and its variants essentially measure the closeness of the initial prior hypothesis to the target optimal hypothesis by the needed minibatch gradient steps to move from the former to the latter. More specifically, MAML seeks to find a good initialization $\boldsymbol{w}$ such that $\boldsymbol{w}_T^* = \boldsymbol{w} - \eta\nabla\mathcal{L}_{D_T}(\boldsymbol{w}) = \arg\min_{\boldsymbol{w}_T}\langle\nabla\mathcal{L}_{D_T}(\boldsymbol{w}), \boldsymbol{w}_T - \boldsymbol{w}\rangle + \frac{1}{2\eta}\|\boldsymbol{w}_T - \boldsymbol{w}\|_2^2$ would be close to the optimal hypothesis of task $T$. In contrast, Meta-MinibatchProx finds the task-specific optimal hypothesis through minibatch proxmal update (1), namely $\boldsymbol{w}_T^* = \min_{\boldsymbol{w}_T} \mathcal{L}_{D_T}(\boldsymbol{w}_T) + \frac{\lambda}{2}\|\boldsymbol{w}_T - \boldsymbol{w}\|_2^2$. In comparison, one can observe that MAML actually approximates the loss $\mathcal{L}_{D_T}(\boldsymbol{w}_T)$ by its first-order taylor expansion for minibatch proximal update, while Meta-MinibatchProx directly optimizes $\mathcal{L}_{D_T}(\boldsymbol{w}_T) = \langle\nabla\mathcal{L}_{D_T}(\boldsymbol{w}), \boldsymbol{w}_T - \boldsymbol{w}\rangle + \frac{1}{2}\langle\nabla^2\mathcal{L}_{D_T}(\boldsymbol{w})(\boldsymbol{w}_T - \boldsymbol{w}), (\boldsymbol{w}_T - \boldsymbol{w})\rangle + \frac{1}{6}\langle\nabla^3\mathcal{L}_{D_T}(\boldsymbol{w}), (\boldsymbol{w}_T - \boldsymbol{w})^{\otimes^3}\rangle + \cdots$ and thus can make use of higher-order information of $\mathcal{L}_{D_T}$ beyond gradient to search the optimal hypothesis around $\boldsymbol{w}$. In this way, Meta-MinibatchProx is able to find better task-specific hypotheses which in turn leads to more accurate estimation of the prior hypothesis. As we will show shortly that such a minibatch proximal update scheme turns out to be more suitable for algorithm implementation, generalization analysis and it works reasonably well for few-shot learning tasks in the experiments.

As another advantage of Meta-MinibatchProx over MAML, it can be readily modified to handle outlier meta-tasks by using certain robust variants of the $\ell_{p,q}$-norm regularizer $\|\boldsymbol{w}_{T_i} - \boldsymbol{w}\|_p^q$ for minibatch proximal update. For instance, suppose that there are a few outlier-tasks $\mathcal{O} = \{T_o\}$ whose optima $\{\boldsymbol{w}_o\}$ are quite far from the optima $\{\boldsymbol{w}_s\}$ of the inlier(normal)-tasks $\mathcal{S} = \{T_s\}$. To handle this case, Meta-MinibatchProx may adopt the robust $\ell_{2,1}$-norm $\frac{1}{n}\sum_{i=1}^n \|\boldsymbol{w}_{T_i} - \boldsymbol{w}\|_2$ which can tolerate relatively large distances between $\boldsymbol{w}$ and $\{\boldsymbol{w}_o\}$ [37, 38], while the learned prior $\boldsymbol{w}^*$ is still close to the optima $\{\boldsymbol{w}_s\}$ and only requires a few training data for adaptation to new inlier-tasks. In contrast, it is hard to tailor MAML to a robust version due to its fixed update rule which is less flexible to be adapted to handle outlier-tasks. As a result, the meta-initialization $\boldsymbol{w}^*$ returned by MAML is expected to departure far away from $\{\boldsymbol{w}_s\}$ and thus needs more data for adaptation to new inlier-tasks. Experimental results in Sec. 4.3 well demonstrate the advantages of Meta-MinibatchProx over MAML in such a regime of robust meta-learning.

## 3.2 Stochastic Gradient Meta-Optimization

Here we propose an SGD based meta-optimization algorithm to solve the min-min problem (3). To develop the algorithm, we first establish the following simple lemma to show that the gradient of the meta-loss $\phi_{D_T}(\boldsymbol{w}) = \min_{\boldsymbol{w}_T} \mathcal{L}_{D_T}(\boldsymbol{w}_T) + \frac{\lambda}{2}\|\boldsymbol{w}_T - \boldsymbol{w}\|_2^2$ can be expressed in closed-form based on the optimizer of the associated minibatch proximal update.

**Lemma 1.** *Assume that $\mathcal{L}_{D_T}$ is differentiable and $\boldsymbol{w}_T^*$ is the unique minimizer of $\mathcal{L}_{D_T}(\boldsymbol{w}_T) + \frac{\lambda}{2}\|\boldsymbol{w}_T - \boldsymbol{w}^*\|_2^2$. Then the gradient of the meta-loss $\phi_{D_T}(\boldsymbol{w})$ is given by $\nabla\phi_{D_T}(\boldsymbol{w}) = \lambda(\boldsymbol{w} - \boldsymbol{w}_T^*)$.*

See its proof in Appendix B.2. This lemma shows that the gradient evaluation of meta-loss can be computed via solving the intra-task minibatch proximal update problem. This differs from MAML in which the gradient evaluation of a task-specific meta-loss is relying on the second-order information.

Lemma 1 lays the foundation of our SGD based meta-optimization algorithm as outlined in Algorithm 1. At the $s$-th iteration, we first sample a mini-batch tasks $\{T_i\}$ of size $b_s$ and then perform the intra-task minibatch proximal update to find an approximate optimal hypothesis $\boldsymbol{w}_{T_i}^s$ for each task $T_i$. For implementing this step, we conventionally use the warm-start approach, namely taking the current prior hypothesis parameters $\boldsymbol{w}^s$ as the initialization of SGD to solve the subproblem. According to Lemma 1, the average meta-gradient over the minibatch task is $\lambda(\boldsymbol{w}^s - \frac{1}{b_s}\sum_{i=1}^{b_s}\boldsymbol{w}_{T_i}^{*s})$ where $\boldsymbol{w}_{T_i}^{*s} = \operatorname{argmin}_{\boldsymbol{w}_{T_i}} \mathcal{L}_{D_{T_i}}(\boldsymbol{w}_{T_i}) + \frac{\lambda}{2}\|\boldsymbol{w}_{T_i} - \boldsymbol{w}^s\|_2^2$. Since the exact minimizer $\boldsymbol{w}_{T_i}^{*s}$ is generally hard to estimate, we alternatively use the sub-optimal solution $\boldsymbol{w}_{T_i}^s$ to approximate $\boldsymbol{w}_{T_i}^{*s}$ and update the meta-parameter via minibatch SGD with learning rate $\eta_s$: $\boldsymbol{w}^{s+1} = \boldsymbol{w}^s - \eta_s\lambda(\boldsymbol{w}^s - \frac{1}{b_s}\sum_{i=1}^{b_s}\boldsymbol{w}_{T_i}^s)$. As a first-order meta-optimization method without accessing the Hessian matrix of intra-task empirical risk function, Algorithm 1 is expected to be more efficient in computation and memory for large-scale problems. Due to the independence of intra-task minibatch proximal update, the meta-gradient evaluation step can be easily parallelized and distributed to accelerate the training process.

Before analyzing Algorithm 1, we first give some definitions conventionally used in machine learning.

**Definition 1** (Convexity, Lipschitz continuity, and Smoothness). *We say a function $g(\boldsymbol{w})$ is $\mu$-strongly convex if $\forall \boldsymbol{w}_1, \boldsymbol{w}_2, g(\boldsymbol{w}_1) \geq g(\boldsymbol{w}_2) + \langle\nabla g(\boldsymbol{w}_2), \boldsymbol{w}_1 - \boldsymbol{w}_2\rangle + \frac{\mu}{2}\|\boldsymbol{w}_1 - \boldsymbol{w}_2\|_2^2$. If $\mu = 0$, then we say $g(\boldsymbol{w})$ is convex. Moreover, we say $g(\boldsymbol{w})$ is G-Lipschitz continuous if $\|g(\boldsymbol{w}_1) - g(\boldsymbol{w}_2)\|_2 \leq G\|\boldsymbol{w}_1 - \boldsymbol{w}_2\|_2$ with a universal constant G. $g(\boldsymbol{w})$ is said to be L-smooth if its gradient obeys $\|\nabla g(\boldsymbol{w}_1) - \nabla g(\boldsymbol{w}_2)\|_2 \leq L\|\boldsymbol{w}_1 - \boldsymbol{w}_2\|_2$ with a universal constant L.*

The following theorem summaries our main results on the convergence of Algorithm 1.

**Theorem 1.** *Suppose each loss $\mathcal{L}_{D_T}(\boldsymbol{w}_T)$ is differentiable and for each task, its optimum $\boldsymbol{w}_T^* = \operatorname{argmin}_{\boldsymbol{w}_T} \mathcal{L}_{D_T}(\boldsymbol{w}_T) + \frac{\lambda}{2}\|\boldsymbol{w}_T - \boldsymbol{w}\|_2^2$ satisfies $\mathbb{E}[\|\boldsymbol{w}_T^* - \boldsymbol{w}\|_2^2] \leq \sigma^2$. Let $\boldsymbol{w}^* = \operatorname{argmin}_{\boldsymbol{w}} F(\boldsymbol{w})$ and $\boldsymbol{w}_{T_i}^{*s} = \operatorname{argmin}_{\boldsymbol{w}_{T_i}} \mathcal{L}_{D_{T_i}}(\boldsymbol{w}_{T_i}) + \frac{\lambda}{2}\|\boldsymbol{w}_{T_i} - \boldsymbol{w}^s\|_2^2$ is the optimal parameters to task $T_i$.*

*(1) Convex setting. Assume $\mathcal{L}_{D_T}(\boldsymbol{w}_T)$ is convex. Then by setting $\eta_s = \frac{2}{s\lambda}$, $\epsilon_s = \frac{c}{S}$, $\alpha = \frac{8S\lambda^2\sigma^2}{S-1} + c(1 + \frac{8}{S-1}))$ with a constant c, we have*

$$\mathbb{E}[\|\boldsymbol{w}^S - \boldsymbol{w}^*\|_2^2] \leq \frac{\alpha}{\lambda^2 S} \qquad and \qquad \mathbb{E}\Big[\Big\|\frac{1}{n}\sum_{i=1}^n \boldsymbol{w}_{T_i}^{*S} - \boldsymbol{w}^S\Big\|_2^2\Big] \leq \frac{L^2\alpha}{(\lambda+L)^2 S}.$$

*(2) Non-convex setting. Assume $\mathcal{L}_{D_T}(\boldsymbol{w}_T)$ is L-smooth. Then by setting $\lambda > L$, $\eta_s = \sqrt{\frac{\Delta}{\gamma S}}$, $\epsilon_s = \frac{c}{\sqrt{S}}$ with $\gamma = \frac{\lambda^3 L}{(\lambda+L)}\big(\sigma^2 + \frac{c}{\sqrt{S}(\lambda-L)^2}\big)$ and $\Delta = F(\boldsymbol{w}^0) - F(\boldsymbol{w}^*)$, we have*

$$\min_s \mathbb{E}[\|\nabla F(\boldsymbol{w}^s)\|_2^2] = \lambda^2 \min_s \mathbb{E}\Big[\Big\|\frac{1}{n}\sum_{i=1}^n \boldsymbol{w}_{T_i}^{*s} - \boldsymbol{w}^s\Big\|_2^2\Big] \leq \frac{1}{\sqrt{S}}\Big[4\sqrt{\Delta\gamma} + \frac{2c\lambda^2}{(\lambda-L)^2}\Big].$$

See Appendix B.3 for its proof. The assumptions in Theorem 1 are standard in stochastic optimization [39, 40, 41, 42]. The theorem guarantees that Algorithm 1 can converge for both convex and non-convex loss function $\mathcal{L}_{D_T}(\boldsymbol{w}_T)$. Specifically, for convex loss $\mathcal{L}_{D_T}(\boldsymbol{w}_T)$, the convergence rate of Algorithm 1 is at the order of $\mathcal{O}(\frac{1}{S})$, while for non-convex case, the convergence rate is of the order $\mathcal{O}(\frac{1}{\sqrt{S}})$. Besides, we further prove the distance $\|\frac{1}{n}\sum_{i=1}^n \boldsymbol{w}_{T_i}^{*S} - \boldsymbol{w}^S\|_2$ will be very small in expectation after sufficient training iterations. This means that the computed initialization $\boldsymbol{w}^S$ will

be very close in average to the optimal hypothesis $\boldsymbol{w}_{T_i}^{*S}$ of task $T_i$ drawn from the observed $n$ tasks. As the $n$ tasks are sampled from task distribution $\mathcal{T}$, the prior hypothesis meta-regularizer $\boldsymbol{w}^S$ is expected to have small distance to the optimal hypothesis $\boldsymbol{w}_T^{*S}$ of task $T$ draw from $\mathcal{T}$. Intuitively speaking, this result justifies that the meta-regularizer $\boldsymbol{w}^S$ is close to the desired hypothesis of each task and thus serves as a good regularizer in the task-specific minibatch proximal update.

More generally, we actually can show the asymptotic convergence of Algorithm 1 if the learning rate obeys $\eta_s < \frac{2}{\lambda}$ and $F(\boldsymbol{w})$ is lower bounded, namely, $\inf_{\boldsymbol{w}} F(\boldsymbol{w}) > -\infty$. Specifically, Theorem 4 in Appendix A.1 guarantees that 1) the sequence $\{\boldsymbol{w}^s\}$ produced by Algorithm 1 can decrease the loss function $F(\boldsymbol{w})$ monotonically and 2) the accumulation point $\boldsymbol{w}^*$ to the sequence $\{\boldsymbol{w}^s\}$ converges to a Karush–Kuhn–Tucker point of problem (3), which guarantees the convergence performance of the proposed algorithm. Such results still hold when the loss $\mathcal{L}_{D_T}(\boldsymbol{w}_T)$ is non-smooth, e.g. involving hinge loss and/or $\ell_1$-norm regularization. Prior optimization based meta-learning approaches, such as MAML [6], FOMAML [6] and Reptile [10], only provide empirical convergence results but lack of provable convergence guarantees as provided in this work.

### 3.3 Statistical Justification: Benefit of Hypothesis Transfer in Meta Learning

We further show how the prior hypothesis transfer can be beneficial to minibatch proximal update for future tasks, which theoretically justifies the advantage of Meta-MinibatchProx for few-shot learning. Assume that we have learned an optimal prior hypothesis $\boldsymbol{w}^* = \operatorname{argmin}_{\boldsymbol{w}} F(\boldsymbol{w})$. For our discussion here, we view $\boldsymbol{w}^*$ as a deterministic hypothesis because the uncertainty associated with $\boldsymbol{w}^*$ does not play a role in the following analysis. Let $T \sim \mathcal{T}$ be any future task from which $K$ samples $D_T = \{(\boldsymbol{x}_i, \boldsymbol{y}_i)\}_{i=1}^K$ are randomly sampled. The minibatch proximal update on $T$ is then given by $\boldsymbol{w}_T^* = \operatorname{argmin}_{\boldsymbol{w}_T} \{\mathcal{L}_{D_T}(\boldsymbol{w}_T) + \frac{\lambda}{2}\|\boldsymbol{w}_T - \boldsymbol{w}^*\|^2\}$. Theorem 2 basically shows the impact of the prior hypothesis $\boldsymbol{w}^*$ on reducing the excess risk of $\boldsymbol{w}_T^*$ when the former is sufficiently close to the optimal population solution in expectation. See its proof in Appendix C.2.

**Theorem 2.** *Suppose $\ell(f(\boldsymbol{w}, \boldsymbol{x}), \boldsymbol{y})$ is G-Lipschitz continuous, L-smooth and convex w.r.t. $\boldsymbol{w}$. For any $T \sim \mathcal{T}$ and $D_T = \{(\boldsymbol{x}_i, \boldsymbol{y}_i)\}_{i=1}^K \sim T$, we respectively let $\boldsymbol{w}_{T,E}^* \in \operatorname{argmin}_{\boldsymbol{w}_T} \{\mathcal{L}(\boldsymbol{w}_T) := \mathbb{E}_{(\boldsymbol{x}, \boldsymbol{y}) \sim T} [\ell(f(\boldsymbol{w}_T, \boldsymbol{x}), \boldsymbol{y})]\}$ and $\boldsymbol{w}_T^* = \operatorname{argmin}_{\boldsymbol{w}_T} \mathcal{L}_{D_T}(\boldsymbol{w}_T) + \frac{\lambda}{2}\|\boldsymbol{w}_T - \boldsymbol{w}^*\|_2^2$. Then we have*

$$\mathbb{E}_{T \sim \mathcal{T}} \mathbb{E}_{D_T \sim T} [\mathcal{L}(\boldsymbol{w}_T^*) - \mathcal{L}(\boldsymbol{w}_{T,E}^*)] \leq \frac{4G^2}{\lambda K} + \frac{\lambda}{2} \mathbb{E}_{T \sim \mathcal{T}} [\|\boldsymbol{w}^* - \boldsymbol{w}_{T,E}^*\|^2].$$

Theorem 2 shows that for convex loss $\ell(f(\boldsymbol{w}, \boldsymbol{x}), \boldsymbol{y})$, the excess risk of the output hypothesis $\boldsymbol{w}_T^*$ on the task $T$ via minibatch proximal update is decided by two factors, i.e., the training sample number $K$ for each task $T \sim \mathcal{T}$ and the expected distance $\mathbb{E}_{T \sim \mathcal{T}} [\|\boldsymbol{w}^* - \boldsymbol{w}_{T,E}^*\|^2]$ between the meta-regularizer $\boldsymbol{w}^*$ provided by Meta-MinibatchProx and the optimal population hypothesis $\boldsymbol{w}_{T,E}^*$ for task $T$. Specifically, if $K$ increases, then the first term in the upper bound becomes smaller. Moreover, the closer $\boldsymbol{w}^*$ is to $\boldsymbol{w}_{T,E}^*$, the better the updated hypothesis $\boldsymbol{w}_T^*$ approaches to $\boldsymbol{w}_{T,E}^*$ and thus enjoys better generalization performance for a new task drawn from task distribution $\mathcal{T}$ in expectation. Indeed, by choosing proper value of $\lambda$, we can balance the two terms in the above excess risk bound. For instance, by letting $\lambda = \sqrt{8G^2/(K \mathbb{E}_{T \sim \mathcal{T}}[\|\boldsymbol{w}^* - \boldsymbol{w}_{T,E}^*\|_2^2])}$, then the expected excess risk $\mathbb{E}_{T \sim \mathcal{T}} \mathbb{E}_{D_T} [\mathcal{L}(\boldsymbol{w}_T^*) - \mathcal{L}(\boldsymbol{w}_{T,E}^*)]$ is at the order of $\mathcal{O}\left(\frac{1}{\sqrt{K}} \sqrt{\mathbb{E}_{T \sim \mathcal{T}}[\|\boldsymbol{w}^* - \boldsymbol{w}_{T,E}^*\|_2^2]}\right)$. These results justify the benefit of hypothesis transfer in Meta-MinibatchProx. For non-convex loss $\ell(f(\boldsymbol{w}, \boldsymbol{x}), \boldsymbol{y})$, Theorem 5 in Appendix A.2 also provides excess risk analysis which shows very similar roles of the training sample number $K$ and the distance $\mathbb{E}_{T \sim \mathcal{T}} [\|\boldsymbol{w}^* - \boldsymbol{w}_{T,E}^*\|^2]$ as those in Theorem 2.

For non-convex loss, we have an additional result on the first-order optimality which is formally stated in Theorem 3. See its proof in Appendix C.3.

**Theorem 3.** *Suppose $\ell(f(\boldsymbol{w}, \boldsymbol{x}), \boldsymbol{y})$ is G-Lipschitz continuous and L-smooth w.r.t. $\boldsymbol{w}$. For any $T \sim \mathcal{T}$ and $D_T = \{(\boldsymbol{x}_i, \boldsymbol{y}_i)\}_{i=1}^K \sim T$, we let $\boldsymbol{w}_{T,E}^* \in \operatorname{argmin}_{\boldsymbol{w}_T} \{\mathcal{L}(\boldsymbol{w}_T) := \mathbb{E}_{(\boldsymbol{x}, \boldsymbol{y}) \sim T} [\ell(f(\boldsymbol{w}_T, \boldsymbol{x}), \boldsymbol{y})]\}$ and $\boldsymbol{w}_T^* = \operatorname{argmin}_{\boldsymbol{w}_T} \mathcal{L}_{D_T}(\boldsymbol{w}_T) + \frac{\lambda}{2}\|\boldsymbol{w}_T - \boldsymbol{w}^*\|_2^2$, respectively. Then for $\lambda > L$, it holds that*

$$\mathbb{E}_{T \sim \mathcal{T}} \left[ \|\mathbb{E}_{D_T \sim T} [\nabla \mathcal{L}(\boldsymbol{w}_T^*)]\|^2 \right] \leq \frac{32G^2L^2}{(\lambda - L)^2 K^2} + \frac{8G^2}{(\lambda - L)\beta K} + \frac{2}{\beta} \mathbb{E}_{T \sim \mathcal{T}} [\mathcal{L}(\boldsymbol{w}^*) - \mathcal{L}(\boldsymbol{w}_T^*)],$$

*where $\beta = \frac{1}{\lambda} \left[ 1 - \frac{L}{2\lambda} \right]$.*

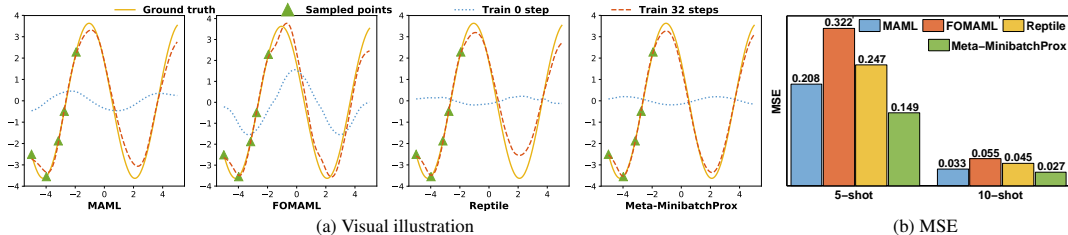

(a) Visual illustration                                             (b) MSE

Figure 1: Comparison on the few-shot regression problem. (a) shows the prediction results of sine wave function when fine-training a meta model on five samples. (b) reports the mean squared errors between the prediction of sine function and the ground truth on 200 testing tasks.

Theorem 3 reveals that the training sample number $K$ and the distance between the expected loss $\mathcal{L}(\boldsymbol{w})$ at the prior hypothesis meta-regularizer $\boldsymbol{w}^*$ (e.g., learnt by Meta-MinibatchProx) and the optimal hypothesis parameter $\boldsymbol{w}_T^*$ of the task $T$ are all critical for obtaining the first-order optimal hypothesis parameter $\boldsymbol{w}_T^*$ in a task $T \sim \mathcal{T}$. Actually, the roles of such two factors in Theorem 3 are consistent with those in Theorem 2. Specifically, the more training samples we have, the smaller of the gradient at the bias hypothesis $\boldsymbol{w}_T^*$, which means $\boldsymbol{w}_T^*$ is close to a stationary hypothesis $\boldsymbol{w}_{T,E}^*$ of the expected risk $\mathcal{L}(\boldsymbol{w}_T) = \mathbb{E}_{(\boldsymbol{x},\boldsymbol{y}) \sim T}\left[\ell(f(\boldsymbol{w}_T, \boldsymbol{x}), \boldsymbol{y})\right]$. Moreover, if the provided initialization $\boldsymbol{w}^*$ is close enough to the optimal hypothesis $\boldsymbol{w}_{T,E}^*$, then their corresponding losses on the task $T \sim \mathcal{T}$ should also be close, which in turn implies good first-order optimum hypothesis parameter $\boldsymbol{w}_T^*$.

## 4 Experiments

We present in this section the performance evaluation of our Meta-MinibatchProx method on benchmark few-shot regression and classification tasks with comparison against several representative state-of-the-art meta-learning approaches. The code is available at https://panzhous.github.io.

### 4.1 Results on Regression Tasks

**Experimental setting.** Following [10], here we consider a synthetic one-dimensional sine wave regression problem. The target function is $y(x) = a\sin(x + b)$ where the amplitude $a$ and the phase $b$ are respectively uniformly sampled from the intervals $[0.1, 5.0]$ and $[0, 2\pi]$. Then for each training task, with fixed $a$ and $b$ the learner samples $p$ points $x_1, \cdots, x_p$ uniformly drawn from the intervals $[-5.0, 5.0]$ to fit the whole function $y(x)$. As shown in [10], this problem is instructive, since joint training cannot learn a useful initialization as the average function $\mathbb{E}[y(x)] = 0$ due to the random phase, while meta learning approaches can work well. After learning an initialization, in the testing phase, we randomly sample an amplitude $a$ and a phase $b$ as aforementioned to produce a new task. Then we randomly sample $K$ data points from $[-5.0, 5.0]$ for training and use 200 testing samples evenly distributed on $[-5.0, 5.0]$ to compute the mean squared errors (MSE) between the prediction and the ground truth. We repeat this procedure 200 times and report the average of MSE. Following [10], in the experiments, we set $p = 50$ for training, and respectively set $K = 5$ and $K = 10$ for testing. For the regression network, we adopt a small network with two-hidden layers of size 40 and Tanh nonlinear functions. Here we use Tanh function instead of ReLU because Tanh gives a slightly better performance on all the considered approaches. For our Meta-MinibatchProx, we set $\lambda = 0.5$ and use SGD to solve the inner subproblem with 15 steps of iteration with learning rate 0.02. For the learning rate $\eta_s$ in Meta-MinibatchProx, we decrease it at each iteration as $\eta_s = \alpha(1 - s/S)$ where the total iteration number $S$ in Algorithm 1 and $\alpha$ are set to 30,000 and 0.8, respectively.

**Results.** From the curves in Fig. 1 (a) we can observe that after training, all the compared meta-learning approaches can well infer the amplitude and phase, and thus can predict the entire sine function, although they only see five data points which are all in half of the input range. See Fig. 3 in Appendix D for more visualization results. These results demonstrate that the considered approaches can learn a good model prior hypothesis and thus can quickly adapt to a new task with only a few training samples. Compared with others, Meta-MinibatchProx can better fit the underlying function. Starting from the prior hypothesis, MAML and its variants run several gradient descent steps to find task-specific optimal hypotheses and then use them to update the prior hypothesis. In contrast, through using minibatch proxmal update Meta-MinibatchProx is able to make use of higher-order information of the empirical risk instead of only using the first-order information as in MAML to

Table 1: Few-shot classification accuracy (%) of the compared approaches on the miniImageNet dataset. The reported accuracies are with 95% confidence intervals.

| method | 1-shot 5-way | 5-shot 5-way | 1-shot 20-way | 5-shot 20-way |
|---|---|---|---|---|
| Matching Net [8] | $43.56 \pm 0.84$ | $55.31 \pm 0.73$ | $17.31 \pm 0.22$ | $22.69 \pm 0.20$ |
| Meta-LSTM [5] | $43.33 \pm 0.77$ | $60.60 \pm 0.71$ | $16.70 \pm 0.23$ | $26.06 \pm 0.25$ |
| MAML [6] | $46.21 \pm 1.76$ | $61.12 \pm 1.01$ | $16.01 \pm 0.52$ | $18.34 \pm 0.33$ |
| FOMAML [6] | $45.53 \pm 1.58$ | $61.02 \pm 1.12$ | $15.21 \pm 0.54$ | $17.67 \pm 0.47$ |
| Reptile [10] | $47.07 \pm 0.26$ | $62.74 \pm 0.37$ | $18.27 \pm 0.16$ | $28.71 \pm 0.19$ |
| Meta-MinibatchProx | $\mathbf{48.51 \pm 0.92}$ | $\mathbf{64.15 \pm 0.92}$ | $\mathbf{20.50 \pm 0.35}$ | $\mathbf{33.61 \pm 0.41}$ |
| MAML + Transduction [6] | $48.70 \pm 1.84$ | $63.11 \pm 0.92$ | $16.49 \pm 0.58$ | $19.29 \pm 0.29$ |
| FOMAML + Transduction [6] | $48.07 \pm 1.75$ | $63.15 \pm 0.91$ | $15.80 \pm 0.61$ | $18.15 \pm 0.43$ |
| Reptile + Transduction [10] | $49.97 \pm 0.32$ | $65.99 \pm 0.58$ | $18.76 \pm 0.17$ | $29.15 \pm 0.22$ |
| Meta-MinibatchProx + Transduction | $\mathbf{50.77 \pm 0.90}$ | $\mathbf{67.43 \pm 0.89}$ | $\mathbf{21.17 \pm 0.38}$ | $\mathbf{34.30 \pm 0.41}$ |

guide the search of task-specific optimal hypothesises around the learned hypothesis, which may lead to better task-specific hypothesises and thus better prior hypothesis. We can also see performance degradation of the first-order variants (FOMAML and Reptile) of MAML, which could be attributed to the information loss caused by gradient approximation in these first-order variants. In Fig. 1 (b), we report the average MSE on 200 independent experiments to measure their overall prediction performance with $K = 5$ and $K = 10$ training points. These numerical results confirm that Meta-MinibatchProx achieves the best prediction performance which is consistent with the visualization results in Fig. 1 (a).

## 4.2 Results on Classification Tasks

**Datasets.** In this experiment we compare our method with several state-of-the-art approaches for few-shot classification on two benchmark datasets of miniImageNet [5] and tieredImageNet [43]. The miniImageNet consists of 100 classes from ImageNet [44] and each class contains 600 images of size $84 \times 84 \times 3$. Following [6, 10], we use the split proposed in [5], which consists of 64 classes for training, 16 classes for validation and the remaining 20 classes for testing. The tieredImageNet dataset contains 608 classes from the ILSVRC-12 dataset [45] and each image is scaled to $84 \times 84 \times 3$. Moreover, tieredImageNet groups classes into broader hierarchy categories corresponding to higher-level nodes in the ImageNet [46]. Specifically, its top hierarchy has 34 categories and they are further split into 20 training categories (351 classes), 6 validation categories (97 classes) and 8 test categories (160 classes). Such a hierarchy structure ensures that all of the training classes are sufficiently distinct from the testing classes, providing a more realistic few-shot learning scenario.

**Experimental setting.** Following [6, 10, 16], in $K$-shot $N$-way few-shot learning task, we adopt the episodic training procedure. More concretely, we randomly sample $N$ classes from the training classes in a testing dataset and then for each class we randomly draw $K + 1$ instances. The first $K$ instances are for training and the remaining one is for testing. For fairness, like [6, 10], we use a convolution network with 4 modules, in which each module consists of $3 \times 3$ convolutions, followed by batch normalization, $2 \times 2$ max-pooling and a ReLU activation layer. Moreover, for each convolution module, its filter number is 32. We use the same network architecture for both datasets.

In Meta-MinibatchProx, its regularization constant $\lambda$ is set to be $0.1$ for 5-way problem in miniImageNet, and 10 for all the remaining experiments. The robustness of Meta-MinibatchProx to the choice of $\lambda$ is shown in Fig. 2 in Appendix D. We use Adam [47] to solve the inner subproblem with learning rate $1e-3$ for both datasets. The Adam step number for the inner loop is set to 8 for 5-way problems in miniImageNet and 16 for all remaining testing, which are sufficient to compute a good approximate solution for each task due to a few training data. For the learning rate $\eta_s$ in Meta-MinibatchProx, like the regression task, we also decrease it as $\eta_s = \alpha(1 - s/S)$ with $S = 10,0000$, where we set $\alpha$ as $0.1$ for 20 way problem in miniImageNet and 1 in the remaining testing. We test Meta-MinibatchProx on 2,000 episodes and report the average result with 95% confidence intervals. Like [6, 10], we evaluate the testing methods under both transduction and non-transduction settings. For transduction, the information was shared between the test data via batch normalization, while in non-transduction setting, batch normalization statistics are collected from all training samples and a single test sample.

**Results.** We respectively report the classification accuracy results on miniImageNet and tieredImageNet in Table 1 and 2. From these results, one can observe that Meta-MinibatchProx consistently outperforms the existing optimization based methods, including MAML, FOMAML, Reptile and

Table 2: Few-shot classification accuracy (%) of the compared approaches on the tieredImageNet dataset. The reported accuracies are with 95% confidence intervals.

| method | 1-shot 5-way | 5-shot 5-way | 1-shot 10-way | 5-shot 10-way |
| --- | --- | --- | --- | --- |
| Matching Net [8] | $34.95 \pm 0.89$ | $43.95 \pm 0.85$ | $22.46 \pm 0.34$ | $31.19 \pm 0.30$ |
| Meta-LSTM [5] | $33.71 \pm 0.76$ | $46.56 \pm 0.79$ | $22.09 \pm 0.43$ | $35.65 \pm 0.39$ |
| MAML [6] | $49.60 \pm 1.83$ | $66.58 \pm 1.78$ | $33.18 \pm 1.23$ | $49.05 \pm 1.32$ |
| FOMAML [6] | $48.01 \pm 1.74$ | $64.07 \pm 1.72$ | $30.31 \pm 1.12$ | $46.54 \pm 1.24$ |
| Reptile [10] | $49.12 \pm 0.43$ | $65.99 \pm 0.42$ | $31.79 \pm 0.28$ | $47.82 \pm 0.30$ |
| Meta-MinibatchProx | $\mathbf{50.14 \pm 0.92}$ | $\mathbf{68.30 \pm 0.91}$ | $\mathbf{33.68 \pm 0.64}$ | $\mathbf{51.84 \pm 0.65}$ |
| MAML + Transduction [6] | $51.67 \pm 1.81$ | $70.30 \pm 1.75$ | $34.44 \pm 1.19$ | $53.32 \pm 1.33$ |
| FOMAML + Transduction [6] | $50.12 \pm 1.82$ | $67.43 \pm 1.80$ | $31.53 \pm 1.08$ | $49.99 \pm 1.36$ |
| Reptile + Transduction [10] | $51.06 \pm 0.45$ | $69.94 \pm 0.42$ | $33.79 \pm 0.29$ | $51.27 \pm 0.31$ |
| Meta-MinibatchProx + Transduction | $\mathbf{54.37 \pm 0.93}$ | $\mathbf{71.45 \pm 0.94}$ | $\mathbf{35.56 \pm 0.60}$ | $\mathbf{54.50 \pm 0.71}$ |

Meta-LSTM, as well as metric based approach, namely Matching Net. Specifically, on miniImageNet, Meta-MinibatchProx respectively makes about $1.44\%$, $1.41\%$, $2.23\%$ and $4.90\%$ improvements on the four testing cases (from left to right) under non-transduction setting, and under transduction setting it also brings about $0.80\%$, $1.44\%$, $2.41\%$ and $5.25\%$ improvements for the four cases. Similarly, on tieredImageNet, Meta-MinibatchProx averagely improves by about $1.39\%$ on the four testing cases in the non-transduction setting, and makes $1.54\%$ average improvement on the four testing cases when using transduction technique. These results demonstrate the advantages of Meta-MinibatchProx behind which the potential reasons have been discussed in Sec. 4.1. Besides, by comparing the results of MAML with its first-order variants (FOMAML and Reptile) on tieredImageNet, we can also observe the generalization performance degeneration of the first-order variants. FOMAML directly ignores the second-order derivative and leads to about $2\%$ degeneration in most cases. Reptile approximates the gradient estimation in MAML which also brings information loss and hence suffers from performance degeneration. In contrast, our model can be efficiently optimized via only accessing the first-order information of loss functions without doing any model approximation. The observed outstanding generalization performance of Meta-MinibatchProx also confirms our theory in Sec. 3.3.

### 4.3 Results on Outlier-Corrupted Tasks

We further test a noisy case with the presence of outlier-tasks as described in Sec. 3.1. To do so, we add $5\%$ outlier images with zero pixels into each training class in miniImageNet. If the sampled task $T$ consists of these outlier images, then it forms an outlier-task. For training, similar to Lemma 1, we can compute the gradient of the meta-loss $\phi_{D_T}(\boldsymbol{w})$ as $\nabla\phi_{D_T}(\boldsymbol{w}) = \frac{\lambda(\boldsymbol{w}-\boldsymbol{w}_T^*)}{2\|\boldsymbol{w}-\boldsymbol{w}_T^*\|_2}$, where $\boldsymbol{w}_T^* = \operatorname{argmin}_{\boldsymbol{w}_T} \mathcal{L}_{D_T}(\boldsymbol{w}_T) + \frac{\lambda}{2}\|\boldsymbol{w}_T - \boldsymbol{w}\|_2$. However, since $\|\boldsymbol{w}_T - \boldsymbol{w}\|_2$ is usually very small in practice which makes the algorithm numerically unstable, we choose to approximate this quantity

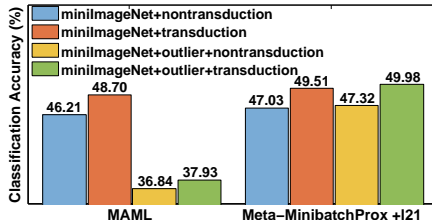

Figure 2: Evaluation with outlier-tasks.

as $\log(1 + \|\boldsymbol{w}_T - \boldsymbol{w}\|_2^2)$ with meta gradient $\nabla\phi_{D_T}(\boldsymbol{w}) = \frac{\lambda(\boldsymbol{w}-\boldsymbol{w}_T^*)}{1+\|\boldsymbol{w}-\boldsymbol{w}_T^*\|_2^2}$. The same experimental protocol as in Sec. 4.2 is used for evaluation and the results are shown in Fig. 2. From this group of results we can observe that Meta-MinibatchProx with $\ell_{21}$-norm regularization achieves substantially better performance than MAML in the considered outlier-corrupted setting, which conforms the flexibility of Meta-MinibatchProx to handle noisy meta-learning problems.

## 5 Conclusion

In this work, we propose Meta-MinibatchProx as a minibatch proximal update based method for learning to hypothesis transfer. The proposed approach seeks to learn from a set of training tasks a prior hypothesis regularized by which minibatch risk minimization can quickly converge to the optimal hypothesis of each training task. For meta-optimization, we develop a scalable stochastic gradient descent algorithm with provable convergence guarantees for a wide range of convex and non-convex learning problems. Theoretically, we justify the benefit of hypothesis transfer to future learning with a few training samples. Extensive experimental results on benchmark datasets demonstrate the superiority of Meta-MinibatchProx over the state-of-the-art meta learning methods.

## Acknowledgements

Xiao-Tong Yuan was supported by National Major Project of China for New Generation of AI (No. 2018AAA0100400) and Natural Science Foundation of China (NSFC) under Grant 61876090 and Grant 61936005. Jiashi Feng was partially supported by NUS IDS R-263- 000-C67-646, ECRA R-263-000-C87-133, MOE Tier-II R-263-000-D17-112 and AI.SG R-263-000-D97-490.

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
