[Supplementary Material]

# Supplementary File for Efficient Meta Learning via Minibatch Proximal Update

**Pan Zhou**[*]  **Xiao-Tong Yuan**[†]  **Huan Xu**[‡]  **Shuicheng Yan**[△]  **Jiashi Feng**[*]
[*] Learning & Vision Lab, National University of Singapore, Singapore
[†] B-DAT Lab, Nanjing University of Information Science & Technology, Nanjing, China
[‡] Alibaba and Georgia Institute of Technology, USA
[△] YITU Technology, Shanghai, China
pzhou@u.nus.edu  xtyuan@nuist.edu.cn  Huan.xu@alibaba-inc.com  {eleyans, elefjia}@nus.edu.sg

## Abstract

This supplementary document contains the technical proofs of convergence results and some additional numerical results of the NIPS'19 submission entitled "Efficient Meta Learning via Minibach Proximal Update". It is structured as follows. Appendix A shows the convergence of Algorithm 1 under very general assumptions in Theorem 4 and also provides the excess risk analysis of the hypothesis transfer in meta learning when the loss $\ell(f(\boldsymbol{w}, \boldsymbol{x}), \boldsymbol{y})$ is non-convex in Theorem 5. Then Appendix B gives the proofs of the main results in Sec. 3.2 including Lemma 1 and Theorem 1, and the convergence results in Appendix A , namely, Theorem 4. Next, in Appendix C we presents the proofs of Theorems 2 and 3 in Sec. 3.3 and Theorem 5 in Appendix A. Finally, more experimental results on few-shot regression task are presented in Appendix D.

## A   More Theoretical Results

### A.1   Convergence Results under Very General Assumptions

We actually can show the convergence of Algorithm 1 under very general assumptions. For instance, such results still hold when the loss $\mathcal{L}_{D_T}(\boldsymbol{w}_T)$ is not differentiable and not smooth, e.g. hinge loss or involving $\ell_1$ norm regularization.

**Theorem 4.** *Assume learning rate satisfies $\eta_s < \frac{2}{\lambda}$ and $\boldsymbol{w}_{T_i}^s = \operatorname{argmin}_{\boldsymbol{w}_{T_i}} \mathcal{L}_{D_{T_i}}(\boldsymbol{w}_{T_i}) + \frac{\lambda}{2}\|\boldsymbol{w}_{T_i} - \boldsymbol{w}^s\|_2^2$. Then the sequence $\{\boldsymbol{w}^s\}$ produced by Algorithm 1 satisfies the following two properties.*
*(1) $F(\boldsymbol{w}^s)$ is monotonically decreasing. Actually, it obeys*

$$\mathbb{E}\left[F(\boldsymbol{w}^{s+1}) - F(\boldsymbol{w}^s)\right] \le \frac{\lambda}{2}\left[1 - \frac{2}{\lambda\eta_s}\right]\mathbb{E}\|\boldsymbol{w}^{s+1} - \boldsymbol{w}^s\|_2^2 < 0.$$

*(2) Assume $F(\boldsymbol{w})$ is lower bounded, namely, $\inf_{\boldsymbol{w}} F(\boldsymbol{w}) > -\infty$. Then we have $\lim_{s\to+\infty} \mathbb{E}[\|\boldsymbol{w}^{s+1} - \boldsymbol{w}^s\|_2] = 0$. Besides, in expectation, the accumulation point $\boldsymbol{w}^*$ of the sequence $\{\boldsymbol{w}^s\}$ is a Karush–Kuhn–Tucker point to $F(\boldsymbol{w})$. It also further satisfies $\mathbb{E}[\boldsymbol{w}^*] = \frac{1}{n}\sum_{i=1}^n \boldsymbol{w}_{T_i}^*$ where $\boldsymbol{w}_{T_i}^* = \operatorname{argmin}_{\boldsymbol{w}_{T_i}} \mathcal{L}_{D_{T_i}}(\boldsymbol{w}_{T_i}) + \frac{\lambda}{2}\|\boldsymbol{w}_{T_i} - \boldsymbol{w}^*\|_2^2$.*

See its proof in Appendix B.4. Theorem 4 shows that the sequence $\{\boldsymbol{w}^s\}$ produced by Algorithm 1 can decrease the loss function $F(\boldsymbol{w})$ monotonically. Besides, under the mild condition, we further prove the accumulation point $\boldsymbol{w}^*$ to the sequence $\{\boldsymbol{w}^s\}$ converges to a Karush–Kuhn–Tucker point, which guarantees the convergence performance of the proposed algorithm. The provable result $\mathbb{E}[\boldsymbol{w}^*] = \frac{1}{n}\sum_{i=1}^n \boldsymbol{w}_{T_i}^*$ also indicates $\mathbb{E}[\boldsymbol{w}^*] = \mathbb{E}_{T_i \sim p(\mathcal{T})}\boldsymbol{w}_{T_i}^*$ as the $n$ tasks are sampled from task set $\mathcal{T}$ according to $p(\mathcal{T})$, and thus implies that $\boldsymbol{w}^*$ is close to the desired hypothesis of each task. So

we only requires a few samples to adapt it to a new task drawn from $\mathcal{T}$. Prior optimization based meta learning approaches, such as MAML [1], FOMAML [1] and Reptile [2], only provide empirical convergence results but lack of rigorous convergence guarantees stated in this work.

### A.2 Statistical Justification: Benefit of Hypothesis Transfer in Meta Learning under Non-convex Setting

Here we provide the excess risk analysis of the hypothesis transfer in meta learning when the loss $\ell(f(\boldsymbol{w}, \boldsymbol{x}), \boldsymbol{y})$ is non-convex. This result can show how the prior hypothesis transfer can be beneficial to minibatch proximal update for future tasks, which theoretically justifies the advantage of Meta-MinibatchProx for few-shot learning.

**Theorem 5.** *Suppose $\ell(f(\boldsymbol{w}, \boldsymbol{x}), \boldsymbol{y})$ is G-Lipschitz continuous and L-smooth w.r.t. $\boldsymbol{w}$. For any $T \sim \mathcal{T}$ and $D_T = \{(\boldsymbol{x}_i, \boldsymbol{y}_i)\}_{i=1}^{K} \sim T$, we respectively let $\boldsymbol{w}_{T,E}^* \in \operatorname{argmin}_{\boldsymbol{w}_T} \{\mathcal{L}(\boldsymbol{w}_T) := \mathbb{E}_{(\boldsymbol{x}, \boldsymbol{y}) \sim T} [\ell(f(\boldsymbol{w}_T, \boldsymbol{x}), \boldsymbol{y})]\}$ and $\boldsymbol{w}_T^* = \operatorname{argmin}_{\boldsymbol{w}_T} \mathcal{L}_{D_T}(\boldsymbol{w}_T) + \frac{\lambda}{2} \|\boldsymbol{w}_T - \boldsymbol{w}^*\|_2^2$, where $\mathcal{L}_{D_T}(\boldsymbol{w}_T) = \frac{1}{K} \sum_{(\boldsymbol{x}, \boldsymbol{y}) \in D_T} \ell(f(\boldsymbol{w}_T, \boldsymbol{x}), \boldsymbol{y})$. Then for non-convex $\ell(f(\boldsymbol{w}, \boldsymbol{x}), \boldsymbol{y})$, by setting $\lambda > L$ it holds that*

$$\mathbb{E}_{T \sim \mathcal{T}} \mathbb{E}_{D_T} \left[ \mathcal{L}(\boldsymbol{w}_T^*) - \mathcal{L}(\boldsymbol{w}_{T,E}^*) \right] \leq \frac{4G^2}{(\lambda - L)K} + \frac{\lambda}{2} \mathbb{E}_{T \sim \mathcal{T}} \left[ \|\boldsymbol{w}^* - \boldsymbol{w}_{T,E}^*\|^2 \right].$$

The Proof of Theorem 5 can be found in Appendix C.4. From Theorem 5, one can obverse that two important factors, namely the training sample number $K$ for each task $T \sim \mathcal{T}$ and the expected distance $\mathbb{E}_{T \sim \mathcal{T}} \left[ \|\boldsymbol{w}^* - \boldsymbol{w}_{T,E}^*\|^2 \right]$ between the meta-regularizer $\boldsymbol{w}^*$ provided by Meta-MinibatchProx and the optimal population hypothesis $\boldsymbol{w}_{T,E}^*$ for task $T$. Actually, those two factors play consistent roles in deciding the excess risk as them in Theorem 2 for convex loss $\ell(f(\boldsymbol{w}, \boldsymbol{x}), \boldsymbol{y})$. Specifically, if $K$ increases, then the first term in the upper bound becomes smaller. Moreover, the closer $\boldsymbol{w}^*$ is to $\boldsymbol{w}_{T,E}^*$, the better the updated hypothesis $\boldsymbol{w}_T^*$ approaches to $\boldsymbol{w}_{T,E}^*$ and thus enjoys better generalization performance for a new task drawn from task set $\mathcal{T}$ in expectation.

### A.3 Extension from Finite-sum Setting to Online Setting

Rigorously, as most experiments, e.g. image classification, have finite task number $n$ though $n$ may be large, this work focuses on off-line setting. But all convergence and generalization guarantees in this work also hold under online setting. So Meta-MinibatchProx actually has guarantees under both settings.

We briefly introduce the proof extension from off-line setting to online setting. **For convergence**, the auxiliary lemmas, e.g. Lemmas $1 \sim 4$, hold for both settings, as they provide certain results for each task loss $\mathcal{L}_{D_{T_i}}(\boldsymbol{w}_{T_i})$ and do not involve off-line and online settings. Let $\phi_{D_{T_i}}(\boldsymbol{w}) = \min_{\boldsymbol{w}_{T_i}} \mathcal{L}_{D_{T_i}}(\boldsymbol{w}_{T_i}) + \frac{\lambda}{2} \|\boldsymbol{w}_{T_i} - \boldsymbol{w}\|_2^2$ and $\boldsymbol{w}_{T_i}^* = \operatorname{argmin}_{\boldsymbol{w}_{T_i}} \mathcal{L}_{D_{T_i}}(\boldsymbol{w}_{T_i}) + \frac{\lambda}{2} \|\boldsymbol{w}_{T_i} - \boldsymbol{w}\|_2^2$. Then when extending Theorem 1 from off-line setting to online setting, the challenge is to extend (a) $\mathbb{E}[\frac{1}{b_s} \sum_{i=1}^{b_s} \phi_{D_{T_i}}(\boldsymbol{w})] = F(\boldsymbol{w})$ and (b) $\mathbb{E}[\frac{1}{b_s} \sum_{i=1}^{b_s} \nabla \phi_{D_{T_i}}(\boldsymbol{w})] = \nabla F(\boldsymbol{w})$ with $F(\boldsymbol{w}) = \frac{1}{n} \sum_{i=1}^{n} \phi_{D_{T_i}}(\boldsymbol{w})$ under off-line setting to (a) and (b) with $F(\boldsymbol{w}) = \mathbb{E}_{T \sim \mathcal{T}} \phi_{D_T}(\boldsymbol{w})$ for online setting. By sampling mini-batch $\{T_i\}$ as $T_i \sim \mathcal{T}$, then (a) and (b) hold for online setting. As tasks $T_i$, e.g. in image classification, usually have uniform distribution $\mathcal{T}$, we can uniformly sample task $T_i$. The remaining proofs of off-line setting and online setting are the same. Similarly, we extend convergence results in Theorem 4 in Appendix from off-line setting to online setting. **For generalization**, Theorems $2 \sim 4$ still hold for online setting without any changes, as they provide performance of empiric solution on $K$ samples in any task $T \sim \mathcal{T}$ on the expected risk and thus do not involve off-line and online settings.

## B Proof of The Results in Sec. 3.2

### B.1 Auxiliary Lemmas

In this section, we introduce auxiliary lemmas which will be used for proving the results in Sec. 3.2.

**Lemma 2.** *Let the function $h(\boldsymbol{x}, \boldsymbol{y}) : \Omega_1 \times \Omega_2 \mapsto \mathbb{R}$ be $\mu$-strongly convex with respect to its variables $(x, y) \in \Omega_1 \times \Omega_2$ for some $\mu \geq 0$. Then the function*

$$\phi(\boldsymbol{x}) := \min_{\boldsymbol{y} \in \Omega_2} h(\boldsymbol{x}, \boldsymbol{y})$$

*is $\mu$-strongly convex.*

*Proof.* Indeed, give $\theta \in [0, 1]$, $\boldsymbol{x}_1, \boldsymbol{x}_2 \in \Omega_1$

$$\phi(\boldsymbol{x}_1) = \min_{\boldsymbol{y} \in \Omega_2} h(\boldsymbol{x}_1, \boldsymbol{y}) = h(\boldsymbol{x}_1, \boldsymbol{y}_1),$$
$$\phi(\boldsymbol{x}_2) = \min_{\boldsymbol{y} \in \Omega_2} h(\boldsymbol{x}_2, \boldsymbol{y}) = h(\boldsymbol{x}_2, \boldsymbol{y}_2).$$

Let $\boldsymbol{x}_\theta = \theta\boldsymbol{x}_1 + (1-\theta)\boldsymbol{x}_2$ and $\boldsymbol{y}_\theta = \theta\boldsymbol{y}_1 + (1-\theta)\boldsymbol{y}_2$. Since $h$ is $\mu$-strongly convex,

$$h(\boldsymbol{x}_\theta, \boldsymbol{y}_\theta) \leq \theta h(\boldsymbol{x}_1, \boldsymbol{y}_1) + (1-\theta)h(\boldsymbol{x}_2, \boldsymbol{y}_2) - \frac{\mu}{2}\theta(1-\theta)\left(\|\boldsymbol{x}_1 - \boldsymbol{x}_2\|^2 + \|\boldsymbol{y}_1 - \boldsymbol{y}_2\|^2\right)$$

$$= \theta\phi(\boldsymbol{x}_1) + (1-\theta)\phi(\boldsymbol{x}_2) - \frac{\mu}{2}\theta(1-\theta)\left(\|\boldsymbol{x}_1 - \boldsymbol{x}_2\|^2 + \|\boldsymbol{y}_1 - \boldsymbol{y}_2\|^2\right)$$

$$\leq \theta\phi(\boldsymbol{x}_1) + (1-\theta)\phi(\boldsymbol{x}_2) - \frac{\mu}{2}\theta(1-\theta)\|\boldsymbol{x}_1 - \boldsymbol{x}_2\|^2.$$

Hence

$$\phi(\boldsymbol{x}_\theta) = \min_{\boldsymbol{y} \in Q_2} h(\boldsymbol{x}_\theta, \boldsymbol{y}) \leq h(\boldsymbol{x}_\theta, \boldsymbol{y}_\theta) \leq \theta\phi(\boldsymbol{x}_1) + (1-\theta)\phi(\boldsymbol{x}_2) - \frac{\mu}{2}\theta(1-\theta)\|\boldsymbol{x}_1 - \boldsymbol{x}_2\|^2.$$

This shows that $\phi(\boldsymbol{x})$ is also $\mu$-strongly convex. $\qquad\square$

**Lemma 3.** *Assume $g(\boldsymbol{w})$ is $\lambda$-strongly convex. Then we have*

$$\langle \nabla g(\boldsymbol{w}), \boldsymbol{w} - \boldsymbol{w}^* \rangle \geq \lambda \|\boldsymbol{w} - \boldsymbol{w}^*\|^2,$$

*where $\boldsymbol{w}^* = \arg\min_{\boldsymbol{w}} g(\boldsymbol{w})$.*

*Proof.* Firstly, we have for any $\boldsymbol{w}_1$ and $\boldsymbol{w}_2$

$$g(\boldsymbol{w}_1) \geq g(\boldsymbol{w}_2) + \langle \nabla g(\boldsymbol{w}_2), \boldsymbol{w}_1 - \boldsymbol{w}_2 \rangle + \frac{\lambda}{2}\|\boldsymbol{w}_1 - \boldsymbol{w}_2\|^2.$$

Similarly, we have

$$g(\boldsymbol{w}_2) \geq g(\boldsymbol{w}_1) + \langle \nabla g(\boldsymbol{w}_1), \boldsymbol{w}_2 - \boldsymbol{w}_1 \rangle + \frac{\lambda}{2}\|\boldsymbol{w}_1 - \boldsymbol{w}_2\|^2.$$

Combine these two inequalities, we can obtain

$$\langle \nabla g(\boldsymbol{w}_1) - \nabla g(\boldsymbol{w}_2), \boldsymbol{w}_1 - \boldsymbol{w}_2 \rangle \geq \lambda \|\boldsymbol{w}_1 - \boldsymbol{w}_2\|^2.$$

Let $\boldsymbol{w}_2 = \boldsymbol{w}^*$, then $\nabla g(\boldsymbol{w}_2) = 0$. This yields the desired result. The proof is completed. $\qquad\square$

**Lemma 4.** *Assume that each loss $\mathcal{L}_{D_T}(\boldsymbol{w}_T)$ is $L$-smoothness with respect to $\boldsymbol{w}_T$. Then if $\lambda > L$, $\phi_{D_T}(\boldsymbol{w}) = \mathcal{L}_{D_T}(\boldsymbol{w}_T^*) + \frac{\lambda}{2}\|\boldsymbol{w}_T^* - \boldsymbol{w}\|^2$ is $\frac{\lambda L}{\lambda + L}$-smoothness with respect to $\boldsymbol{w}$, where $\boldsymbol{w}_T^* = \arg\min_{\boldsymbol{w}_T} \mathcal{L}_{D_T}(\boldsymbol{w}_T) + \frac{\lambda}{2}\|\boldsymbol{w}_T - \boldsymbol{w}^*\|^2$.*

*Proof.* Since $\mathcal{L}_{D_T}(\boldsymbol{w}_T)$ is differentiable, from the first-order optimality condition we know that

$$\nabla \mathcal{L}_{D_T}(\boldsymbol{w}_T^*) + \lambda(\boldsymbol{w}_T^* - \boldsymbol{w}) = 0.$$

Therefore, we can further obtain

$$\nabla^2 \mathcal{L}_{D_T}(\boldsymbol{w}_T^*)\frac{\partial \boldsymbol{w}_T^*}{\partial \boldsymbol{w}} + \lambda\left(\frac{\partial \boldsymbol{w}_T^*}{\partial \boldsymbol{w}} - \boldsymbol{I}\right) = 0.$$

This implies

$$\frac{\partial \boldsymbol{w}_T^*}{\partial \boldsymbol{w}} = \lambda\left(\nabla^2 \mathcal{L}_{D_T}(\boldsymbol{w}_T^*) + \lambda\boldsymbol{I}\right)^{-1}.$$

From Lemma 1, we have

$$\nabla \phi_{D_T}(\boldsymbol{w}) = \lambda(\boldsymbol{w} - \boldsymbol{w}_T^*).$$

Therefore, we can further have

$$\nabla^2 \phi_{D_T}(\boldsymbol{w}) = \lambda\left(\boldsymbol{I} - \frac{\partial \boldsymbol{w}_T^*}{\partial \boldsymbol{w}}\right) = \lambda\left(\boldsymbol{I} - \lambda\left(\nabla^2 \mathcal{L}_{D_T}(\boldsymbol{w}_T^*) + \lambda\boldsymbol{I}\right)^{-1}\right).$$

Note that $\mathcal{L}_{D_T}(\boldsymbol{w}_T)$ is $L$-smoothness with respect to $\boldsymbol{w}_T$. Then it yields

$$\|\nabla^2 \phi_{D_T}(\boldsymbol{w})\| = \lambda\left\|\left(\boldsymbol{I} - \lambda\left(\nabla^2 \mathcal{L}_{D_T}(\boldsymbol{w}_T^*) + \lambda\boldsymbol{I}\right)^{-1}\right)\right\| \le \frac{\lambda L}{\lambda + L}.$$

The proof is completed. □

## B.2 Proof of Lemma 1

*Proof.* By definition we have $\phi_{D_T}(\boldsymbol{w}) = \mathcal{L}_{D_T}(\boldsymbol{w}_T^*) + \frac{\lambda}{2}\|\boldsymbol{w}_T^* - \boldsymbol{w}\|^2$, where $\boldsymbol{w}_T^* = \arg\min_{\boldsymbol{w}_T} \mathcal{L}_{D_T}(\boldsymbol{w}_T) + \frac{\lambda}{2}\|\boldsymbol{w}_T - \boldsymbol{w}\|^2$. Since $\mathcal{L}_{D_T}(\boldsymbol{w}_T)$ is differentiable, from the first-order optimality condition we know that

$$\nabla \mathcal{L}_{D_T}(\boldsymbol{w}_T^*) + \lambda(\boldsymbol{w}_T^* - \boldsymbol{w}) = 0.$$

From the chain rule we have

$$\nabla \phi_{D_T}(\boldsymbol{w}) = \left(\frac{\partial \boldsymbol{w}_T^*}{\partial \boldsymbol{w}}\right)^\top \nabla \mathcal{L}_{D_T}(\boldsymbol{w}_T^*) + \lambda\left(\boldsymbol{I} - \left(\frac{\partial \boldsymbol{w}_T^*}{\partial \boldsymbol{w}}\right)^\top\right)(\boldsymbol{w} - \boldsymbol{w}_T^*)$$

$$= \lambda(\boldsymbol{w} - \boldsymbol{w}_T^*) + \left(\frac{\partial \boldsymbol{w}_T^*}{\partial \boldsymbol{w}}\right)^\top (\nabla \mathcal{L}_{D_T}(\boldsymbol{w}_T^*) + \lambda(\boldsymbol{w}_T^* - \boldsymbol{w})) = \lambda(\boldsymbol{w} - \boldsymbol{w}_T^*).$$

This proves the desired result. □

## B.3 Proof of Theorem 1

*Proof.* Define $h_{D_T}(\boldsymbol{w}_T, \boldsymbol{w}) = \mathcal{L}_{D_T}(\boldsymbol{w}_T) + \frac{\lambda}{2}\|\boldsymbol{w}_T - \boldsymbol{w}\|^2$. Then $h_{D_T}(\boldsymbol{w}_T, \boldsymbol{w})$ is $\lambda$-strongly convex with respect to $(\boldsymbol{w}_T, \boldsymbol{w})$. It follows immediately from Lemma 2 that $\phi_{D_T}(\boldsymbol{w}) = \mathcal{L}_{D_T}(\boldsymbol{w}_T^*) + \frac{\lambda}{2}\|\boldsymbol{w}_T^* - \boldsymbol{w}\|^2 = \arg\min_{\boldsymbol{w}_T} \mathcal{L}_{D_T}(\boldsymbol{w}_T) + \frac{\lambda}{2}\|\boldsymbol{w}_T - \boldsymbol{w}\|^2$ is also $\lambda$-strongly convex.

Next, we provide the convergence analysis. Before proving the results, we first define $\widehat{\phi}_{D_T}(\boldsymbol{w}, \boldsymbol{w}_T) = \mathcal{L}_{D_T}(\boldsymbol{w}_T) + \frac{\lambda}{2}\|\boldsymbol{w}_T - \boldsymbol{w}\|^2$, $\boldsymbol{w}_T^*$ is the optimum solution to the problem $\boldsymbol{w}_T^* = \arg\min_{\boldsymbol{w}_T} \mathcal{L}_{D_T}(\boldsymbol{w}_T) + \frac{\lambda}{2}\|\boldsymbol{w}_T - \boldsymbol{w}^*\|^2$ and $\widehat{\boldsymbol{w}}_T^*$ is $\epsilon_s$-optimum solution to the problem $\min_{\boldsymbol{w}_T} \mathcal{L}_{D_T}(\boldsymbol{w}_T) + \frac{\lambda}{2}\|\boldsymbol{w}_T - \boldsymbol{w}^*\|^2$, namely $\|\nabla\widehat{\phi}_{D_T}(\boldsymbol{w}, \widehat{\boldsymbol{w}}_T^*)\|^2 \le \epsilon_s$. In this way, we update $\boldsymbol{w}_{s+1} = \boldsymbol{w}_s - \eta_s\lambda(\boldsymbol{w}_s - \frac{1}{b_s}\sum_{i=1}^{b_s}\widehat{\boldsymbol{w}}_{T_i}^*)$.

Then we consider the convex setting. First, $\widehat{\phi}_{D_T}(\boldsymbol{w}, \boldsymbol{w}_T) = \mathcal{L}_{D_T}(\boldsymbol{w}_T) + \frac{\lambda}{2}\|\boldsymbol{w}_T - \boldsymbol{w}\|^2$ is $\lambda$-strongly convex with respect to $\boldsymbol{w}_T$. Then from Lemma 3, we have

$$\lambda\|\widehat{\boldsymbol{w}}_T^* - \boldsymbol{w}_T^*\|^2 \le \langle\nabla\widehat{\phi}_{D_T}(\boldsymbol{w}, \widehat{\boldsymbol{w}}_T^*), \widehat{\boldsymbol{w}}_T^* - \boldsymbol{w}_T^*\rangle \le \|\nabla\widehat{\phi}_{D_T}(\boldsymbol{w}, \widehat{\boldsymbol{w}}_T^*)\| \cdot \|\widehat{\boldsymbol{w}}_T^* - \boldsymbol{w}_T^*\|,$$

which implies

$$\|\widehat{\boldsymbol{w}}_T^* - \boldsymbol{w}_T^*\|^2 \le \frac{1}{\lambda^2}\|\nabla\widehat{\phi}_{D_T}(\boldsymbol{w}, \widehat{\boldsymbol{w}}_T^*)\|^2 \le \frac{\epsilon}{\lambda^2}. \tag{4}$$

Then we consider the term $\mathbb{E}[\|\boldsymbol{w}_s - \frac{1}{b_s}\sum_{i=1}^{b_s}\widehat{\boldsymbol{w}}_{T_i}^*\|^2]$:

$$\mathbb{E}[\|\boldsymbol{w}_s - \frac{1}{b_s}\sum_{i=1}^{b_s}\widehat{\boldsymbol{w}}_{T_i}^*\|^2] = \mathbb{E}[\|\boldsymbol{w}_s - \frac{1}{b_s}\sum_{i=1}^{b_s}(\boldsymbol{w}_{T_i}^* + \widehat{\boldsymbol{w}}_{T_i}^* - \boldsymbol{w}_{T_i}^*)\|^2]$$

$$\le 2\mathbb{E}[\|\boldsymbol{w}_s - \frac{1}{b_s}\sum_{i=1}^{b_s}\boldsymbol{w}_{T_i}^*\|^2 + \frac{1}{b_s}\sum_{i=1}^{b_s}\|\widehat{\boldsymbol{w}}_{T_i}^* - \boldsymbol{w}_{T_i}^*\|^2] \overset{①}{\le} 2\sigma^2 + \frac{2\epsilon_s}{\lambda^2},$$

where ① uses the assumption $\mathbb{E}\|\boldsymbol{w}_s - \boldsymbol{w}_{T_i}^*\|^2 \leq \sigma^2$. Next, we can bound the term $\mathbb{E}\langle \boldsymbol{w}_s - \frac{1}{b_s}\sum_{i=1}^{b_s} \widehat{\boldsymbol{w}}_{T_i}^*, \boldsymbol{w}_s - \boldsymbol{w}^* \rangle$ as follows:

$$\mathbb{E}\langle \boldsymbol{w}_s - \frac{1}{b_s}\sum_{i=1}^{b_s} \widehat{\boldsymbol{w}}_{T_i}^*, \boldsymbol{w}_s - \boldsymbol{w}^* \rangle = \mathbb{E}\langle \boldsymbol{w}_s - \frac{1}{b_s}\sum_{i=1}^{b_s} \boldsymbol{w}_{T_i}^*, \boldsymbol{w}_s - \boldsymbol{w}^* \rangle - \frac{1}{b_s}\sum_{i=1}^{b_s} \mathbb{E}\langle \widehat{\boldsymbol{w}}_{T_i}^* - \boldsymbol{w}_{T_i}^*, \boldsymbol{w}_s - \boldsymbol{w}^* \rangle$$

$$= \mathbb{E}\langle \frac{1}{\lambda}\nabla F(\boldsymbol{w}_s), \boldsymbol{w}_s - \boldsymbol{w}^* \rangle - \frac{1}{b_s}\sum_{i=1}^{b_s} \mathbb{E}\langle \widehat{\boldsymbol{w}}_{T_i}^* - \boldsymbol{w}_{T_i}^*, \boldsymbol{w}_s - \boldsymbol{w}^* \rangle$$

$$\overset{①}{\geq} \mathbb{E}\|\boldsymbol{w}_s - \boldsymbol{w}^*\|^2 - \frac{1}{b_s}\sum_{i=1}^{b_s} \mathbb{E}\langle \widehat{\boldsymbol{w}}_{T_i}^* - \boldsymbol{w}_{T_i}^*, \boldsymbol{w}_s - \boldsymbol{w}^* \rangle$$

$$\geq \mathbb{E}\|\boldsymbol{w}_s - \boldsymbol{w}^*\|^2 - \frac{1}{2b_s}\sum_{i=1}^{b_s} \mathbb{E}(\|\widehat{\boldsymbol{w}}_{T_i}^* - \boldsymbol{w}_{T_i}^*\|^2 + \|\boldsymbol{w}_s - \boldsymbol{w}^*\|^2)$$

$$\geq \frac{1}{2}\mathbb{E}\|\boldsymbol{w}_s - \boldsymbol{w}^*\|^2 - \frac{\epsilon_s}{2\lambda^2},$$

where ① holds, since $\phi_{D_T}(\boldsymbol{w})$ is $\lambda$-strongly convex with respect to $\boldsymbol{w}$ and thus $F(\boldsymbol{w}) = \frac{1}{n}\sum_{i=1}^{n} \mathcal{L}_{D_T}(\boldsymbol{w}_T^*) + \frac{\lambda}{2}\|\boldsymbol{w}_T^* - \boldsymbol{w}\|^2$ is also $\lambda$-strongly convex which gives $\langle \nabla F(\boldsymbol{w}), \boldsymbol{w} - \boldsymbol{w}^* \rangle \geq \lambda\|\boldsymbol{w} - \boldsymbol{w}^*\|^2$ in Lemma 3.

Next, we use the above results to prove the convergence results:

$$\mathbb{E}[\|\boldsymbol{w}_{s+1} - \boldsymbol{w}^*\|^2]$$

$$= \mathbb{E}[\|\boldsymbol{w}_s - \boldsymbol{w}^*\|^2] - 2\eta_s\lambda\mathbb{E}\langle \boldsymbol{w}_s - \frac{1}{b_s}\sum_{i=1}^{b_s} \widehat{\boldsymbol{w}}_{T_i}^*, \boldsymbol{w}_s - \boldsymbol{w}^* \rangle + \eta_s^2\lambda^2\mathbb{E}[\|\boldsymbol{w}_s - \frac{1}{b_s}\sum_{i=1}^{b_s} \widehat{\boldsymbol{w}}_{T_i}^*\|^2]$$

$$\overset{①}{\leq} \mathbb{E}[\|\boldsymbol{w}_s - \boldsymbol{w}^*\|^2] - 2\eta_s\lambda\left[\frac{1}{2}\mathbb{E}\|\boldsymbol{w}_s - \boldsymbol{w}^*\|^2 - \frac{\epsilon_s}{2\lambda^2}\right] + \eta_s^2\lambda^2\left[2\sigma^2 + \frac{2\epsilon_s}{\lambda^2}\right]$$

$$\overset{②}{\leq} (1 - \eta_s\lambda)\mathbb{E}[\|\boldsymbol{w}_s - \boldsymbol{w}^*\|^2] + \frac{\eta_s\epsilon_s}{\lambda} + 2\eta_s^2(\lambda^2\sigma^2 + \epsilon_s),$$

$$\overset{②}{\leq} (1 - \eta_s\lambda)\mathbb{E}[\|\boldsymbol{w}_s - \boldsymbol{w}^*\|^2] + 2\eta_s^2(\lambda^2\sigma^2 + \lambda\epsilon_s) + 4\eta_s\epsilon_s,$$

Then by setting $\eta_s = 2/(s\lambda)$, we can further obtain

$$\mathbb{E}[\|\boldsymbol{w}_{s+1} - \boldsymbol{w}^*\|^2] \leq (1 - 2/s)\mathbb{E}[\|\boldsymbol{w}_s - \boldsymbol{w}^*\|^2] + \frac{8(\lambda^2\sigma^2 + \epsilon_s)}{\lambda^2 s^2} + \frac{2\epsilon_s}{\lambda^2 s}.$$

For brevity, let $a_{s+1} = \mathbb{E}[\|\boldsymbol{w}_{s+1} - \boldsymbol{w}^*\|^2]$, $c = \frac{8(\lambda^2\sigma^2 + \epsilon_s)}{\lambda^2}$ and $d = \frac{2\epsilon_s}{\lambda^2}$. Then we can bound $a_s$ as follows:

$$a_s \leq (1 - \frac{2}{s-1})a_{s-1} + \frac{c}{(s-1)^2} + \frac{d}{s-1} \leq a_1\prod_{i=1}^{s-1}(1 - \frac{2}{i}) + \sum_{i=1}^{s-1}(\frac{c}{i^2} + \frac{d}{i})\prod_{j=i+1}^{s-1}(1 - \frac{2}{j})$$

$$\leq \sum_{i=1}^{s-1}(\frac{c}{i^2} + \frac{d}{i})\frac{(i-1)i}{(s-2)(s-1)} \leq \frac{c}{s-1} + \frac{d}{2}.$$

Therefore, by setting $\epsilon_s = \frac{c}{S}$ where $c$ is a constant, we have

$$\mathbb{E}[\|\boldsymbol{w}_S - \boldsymbol{w}^*\|^2] \leq \frac{8(\lambda^2\sigma^2 + c/S)}{\lambda^2 S} + \frac{c}{\lambda^2 S} = \frac{1}{\lambda^2 S}\left(\frac{8S\lambda^2\sigma^2}{S-1} + c\left(1 + \frac{8}{S-1}\right)\right)$$

Besides, from Lemma 4, we know that if each loss $\mathcal{L}_{D_T}(\boldsymbol{w}_T)$ is $L$-smoothness with respect to $\boldsymbol{w}_T$ and $\lambda > L$, $\phi_{D_T}(\boldsymbol{w}) = \mathcal{L}_{D_T}(\boldsymbol{w}_T^*) + \frac{\lambda}{2}\|\boldsymbol{w}_T^* - \boldsymbol{w}\|^2$ is $\frac{\lambda L}{\lambda + L}$-smoothness with respect to $\boldsymbol{w}$, where $\boldsymbol{w}_T^* = \text{argmin}_{\boldsymbol{w}_T} \mathcal{L}_{D_T}(\boldsymbol{w}_T) + \frac{\lambda}{2}\|\boldsymbol{w}_T - \boldsymbol{w}^*\|^2$. So the loss $F(\boldsymbol{w}) = \frac{1}{n}\sum_{i=1}^{n} \mathcal{L}_{D_T}(\boldsymbol{w}_T^*) + \frac{\lambda}{2}\|\boldsymbol{w}_T^* - \boldsymbol{w}\|^2$ is also $\frac{\lambda L}{\lambda + L}$-smoothness. Therefore, we can establish

$$\|\nabla F(\boldsymbol{w})\| = \|\nabla F(\boldsymbol{w}) - \nabla F(\boldsymbol{w}^*)\| \leq \frac{\lambda L}{\lambda + L}\|\boldsymbol{w} - \boldsymbol{w}^*\|.$$

Therefore, we have

$$\mathbb{E}[\|\nabla F(\boldsymbol{w}^S)\|^2] = \mathbb{E}\left[\left\|\frac{1}{n}\sum_{i=1}^{n}\boldsymbol{w}_{T_i}^S - \boldsymbol{w}^S\right\|^2\right] \leq \frac{\lambda^2 L^2}{(\lambda+L)^2}\mathbb{E}[\|\boldsymbol{w}^S - \boldsymbol{w}^*\|^2]$$

$$\leq \frac{L^2}{(\lambda+L)^2 S}\left(\frac{8S\lambda^2\sigma^2}{S-1} + c\left(1 + \frac{8}{S-1}\right)\right).$$

Now we consider non-convex setting. Firstly, by using smoothness assumption that each loss $\mathcal{L}_{D_T}(\boldsymbol{w}_T)$ is $L$-smoothness with respect to $\boldsymbol{w}_T$ and $\lambda > L$, from Lemma 4, we obtain that $\phi_{D_T}(\boldsymbol{w}) = \mathcal{L}_{D_T}(\boldsymbol{w}_T^*) + \frac{\lambda}{2}\|\boldsymbol{w}_T^* - \boldsymbol{w}\|^2$ is $\frac{\lambda L}{\lambda+L}$-smoothness with respect to $\boldsymbol{w}$, where $\boldsymbol{w}_T^* = \operatorname{argmin}_{\boldsymbol{w}_T}\mathcal{L}_{D_T}(\boldsymbol{w}_T) + \frac{\lambda}{2}\|\boldsymbol{w}_T - \boldsymbol{w}^*\|^2$. So the loss $F(\boldsymbol{w}) = \frac{1}{n}\sum_{i=1}^{n}\mathcal{L}_{D_T}(\boldsymbol{w}_T^*) + \frac{\lambda}{2}\|\boldsymbol{w}_T^* - \boldsymbol{w}\|^2$ is also $\frac{\lambda L}{\lambda+L}$-smoothness.

Since $\mathcal{L}_{D_T}(\boldsymbol{w}_T)$ is $L$-smoothness and $\lambda > L$, then $\widehat{\phi}_{D_T}(\boldsymbol{w}, \boldsymbol{w}_T) = \mathcal{L}_{D_T}(\boldsymbol{w}_T) + \frac{\lambda}{2}\|\boldsymbol{w}_T - \boldsymbol{w}\|^2$ is $(\lambda - L)$-strongly convex. Following proof of Eqn. (4), we can prove

$$\|\widehat{\boldsymbol{w}}_T^* - \boldsymbol{w}_T^*\|^2 \leq \frac{1}{(\lambda - L)^2}\|\nabla\widehat{\phi}_{D_T}(\boldsymbol{w}, \widehat{\boldsymbol{w}}_T^*)\|^2 \leq \frac{\epsilon}{(\lambda - L)^2}.$$

Then we consider the term $\mathbb{E}[\|\boldsymbol{w}_{s+1} - \boldsymbol{w}_s\|^2]$:

$$\mathbb{E}[\|\boldsymbol{w}_{s+1} - \boldsymbol{w}_s\|^2] = \eta_s^2\lambda^2\mathbb{E}[\|\boldsymbol{w}_s - \frac{1}{b_s}\sum_{i=1}^{b_s}(\boldsymbol{w}_{T_i}^* + \widehat{\boldsymbol{w}}_{T_i}^* - \boldsymbol{w}_{T_i}^*)\|^2]$$

$$\leq 2\eta_s^2\lambda^2\mathbb{E}[\|\boldsymbol{w}_s - \frac{1}{b_s}\sum_{i=1}^{b_s}\boldsymbol{w}_{T_i}^*\|^2 + \frac{1}{b_s}\sum_{i=1}^{b_s}\|\widehat{\boldsymbol{w}}_{T_i}^* - \boldsymbol{w}_{T_i}^*\|^2]$$

$$\overset{\textcircled{1}}{\leq} 2\eta_s^2\lambda^2\sigma^2 + \frac{2\eta_s^2\lambda^2\epsilon_s}{(\lambda - L)^2},$$

where $\textcircled{1}$ uses the assumption $\mathbb{E}\|\boldsymbol{w}_s - \boldsymbol{w}_{T_i}^*\|^2 \leq \sigma^2$. Next, we can bound the term $\mathbb{E}\langle\nabla F(\boldsymbol{w}^s), \boldsymbol{w}_{s+1} - \boldsymbol{w}_s\rangle$ as follows:

$$\mathbb{E}\langle\nabla F(\boldsymbol{w}^s), \boldsymbol{w}_{s+1} - \boldsymbol{w}_s\rangle = -\eta_s\lambda\mathbb{E}\langle\boldsymbol{w}_s - \frac{1}{b_s}\sum_{i=1}^{b_s}\widehat{\boldsymbol{w}}_{T_i}^*, \nabla F(\boldsymbol{w}^s)\rangle$$

$$= -\eta_s\lambda\mathbb{E}\langle\boldsymbol{w}_s - \frac{1}{b_s}\sum_{i=1}^{b_s}\boldsymbol{w}_{T_i}^*, \nabla F(\boldsymbol{w}^s)\rangle + \eta_s\lambda\frac{1}{b_s}\sum_{i=1}^{b_s}\mathbb{E}\langle\widehat{\boldsymbol{w}}_{T_i}^* - \boldsymbol{w}_{T_i}^*, \nabla F(\boldsymbol{w}^s)\rangle$$

$$\leq -\eta_s\mathbb{E}\|\nabla F(\boldsymbol{w}_s)\|^2 + \eta_s\frac{1}{2b_s}\sum_{i=1}^{b_s}\mathbb{E}(\lambda^2\|\widehat{\boldsymbol{w}}_{T_i}^* - \boldsymbol{w}_{T_i}^*\|^2 + \|\nabla F(\boldsymbol{w}^s)\|^2)$$

$$\leq -\frac{\eta_s}{2}\mathbb{E}\|\nabla F(\boldsymbol{w}_s)\|^2 + \frac{\eta_s\epsilon_s\lambda^2}{(\lambda - L)^2}.$$

Then, we can obtain

$$\mathbb{E}[F(\boldsymbol{w}^{s+1})]$$

$$\leq \mathbb{E}\left[F(\boldsymbol{w}^s) + \mathbb{E}\langle\nabla F(\boldsymbol{w}^s), \boldsymbol{w}_{s+1} - \boldsymbol{w}_s\rangle + \frac{\lambda L}{2(\lambda+L)}\|\boldsymbol{w}_{s+1} - \boldsymbol{w}_s\|^2\right]$$

$$\leq \mathbb{E}\left[F(\boldsymbol{w}^s) - \frac{\eta_s}{2}\mathbb{E}\|\nabla F(\boldsymbol{w}_s)\|^2 + \frac{\eta_s\epsilon_s\lambda^2}{(\lambda - L)^2} + \frac{\lambda L}{2(\lambda+L)}\left(2\eta_s^2\lambda^2\sigma^2 + \frac{2\eta_s^2\lambda^2\epsilon_s}{(\lambda - L)^2}\right)\right].$$

Therefore, by setting $\eta_s = \eta$, we then rearrange and sum up the above inequality to obtain:

$$\min_s \mathbb{E}[\|\nabla F(\boldsymbol{w}^s)\|^2] \leq \frac{1}{S} \sum_{i=1}^s \mathbb{E}\|\nabla F(\boldsymbol{w}^s)\|^2$$

$$\leq \frac{2}{S\eta} \mathbb{E}\left[F(\boldsymbol{w}^0) - F(\boldsymbol{w}^S)\right] + \frac{2\epsilon_s \lambda^2}{(\lambda - L)^2} + \frac{2L\eta\lambda^3}{(\lambda + L)}\left(\sigma^2 + \frac{\epsilon_s}{(\lambda - L)^2}\right)$$

$$\leq \frac{4\sqrt{\Delta\gamma}}{\sqrt{S}} + \frac{2c\lambda^2}{(\lambda - L)^2\sqrt{S}},$$

where in the last inequality, we let $\eta = \sqrt{\frac{\Delta}{\gamma S}}$, $\gamma = \frac{\lambda^3 L}{(\lambda + L)}\left(\sigma^2 + \frac{\epsilon_s}{(\lambda - L)^2}\right)$, $\Delta = F(\boldsymbol{w}^0) - F(\boldsymbol{w}^*) \geq F(\boldsymbol{w}^0) - F(\boldsymbol{w}^S)$, and $\epsilon_s = c/\sqrt{S}$. Therefore, we have

$$\min_s \mathbb{E}[\|\nabla F(\boldsymbol{w}^s)\|^2] = \lambda^2 \min_s \mathbb{E}\left[\left\|\frac{1}{n}\sum_{i=1}^n \boldsymbol{w}_{T_i}^{*s} - \boldsymbol{w}^s\right\|^2\right] \leq \frac{1}{\sqrt{S}}\left[4\sqrt{\Delta\gamma} + \frac{2c\lambda^2}{(\lambda - L)^2}\right].$$

This completes the proof. $\qquad\square$

## B.4    Proof of Theorem 4

*Proof.* We bound the loss function $F(\boldsymbol{w}^{s+1})$ as follows:

$$F(\boldsymbol{w}^{s+1})$$

$$= \frac{1}{n}\sum_{i=1}^n \min_{\boldsymbol{w}_{T_i}}\left\{\mathcal{L}(T_i, \boldsymbol{w}_{T_i}) + \frac{\lambda}{2}\|\boldsymbol{w}_{T_i} - \boldsymbol{w}^{s+1}\|^2\right\}$$

$$= \frac{1}{n}\sum_{i=1}^n\left[\mathcal{L}(T_i, \boldsymbol{w}_{T_i}^{s+1}) + \frac{\lambda}{2}\|\boldsymbol{w}_{T_i}^{s+1} - \boldsymbol{w}^{s+1}\|^2\right]$$

$$\overset{\text{①}}{\leq} \frac{1}{n}\sum_{i=1}^n\left[\mathcal{L}(T_i, \boldsymbol{w}_{T_i}^{s}) + \frac{\lambda}{2}\|\boldsymbol{w}_{T_i}^{s} - \boldsymbol{w}^{s+1}\|^2\right]$$

$$= \frac{1}{n}\sum_{i=1}^n\left[\mathcal{L}(T_i, \boldsymbol{w}_{T_i}^{s}) + \frac{\lambda}{2}\|\boldsymbol{w}_{T_i}^{s} - \boldsymbol{w}^{s}\|^2\right] + \frac{\lambda}{2n}\sum_{i=1}^n\left[2\langle\boldsymbol{w}^s - \boldsymbol{w}^{s+1}, \boldsymbol{w}_i^s - \boldsymbol{w}^s\rangle + \|\boldsymbol{w}^{s+1} - \boldsymbol{w}^s\|^2\right]$$

$$= F(\boldsymbol{w}^s) + \frac{\lambda}{2n}\sum_{i=1}^n\left[2\langle\boldsymbol{w}^s - \boldsymbol{w}^{s+1}, \boldsymbol{w}_i^s - \boldsymbol{w}^s\rangle + \|\boldsymbol{w}^{s+1} - \boldsymbol{w}^s\|^2\right]$$

where ① holds since $\boldsymbol{w}_i^{s+1}$ is the optimum solution to $\min_{\boldsymbol{w}_{T_i}}\left\{\mathcal{L}(T_i, \boldsymbol{w}_{T_i}) + \frac{\lambda}{2}\|\boldsymbol{w}_{T_i} - \boldsymbol{w}^{s+1}\|^2\right\}$. Next, we take expectation on each side of the above inequality and obtain

$$\mathbb{E}[F(\boldsymbol{w}^{s+1})] \leq \mathbb{E}[F(\boldsymbol{w}^s)] + \frac{\lambda}{2}\left[2\mathbb{E}\langle\boldsymbol{w}^s - \boldsymbol{w}^{s+1}, \frac{1}{n}\sum_{i=1}^n \boldsymbol{w}_i^k - \boldsymbol{w}^s\rangle + \mathbb{E}\|\boldsymbol{w}^{s+1} - \boldsymbol{w}^s\|^2\right]$$

$$\overset{\text{①}}{\leq} \mathbb{E}[F(\boldsymbol{w}^s)] + \frac{\lambda}{2}\left[2\mathbb{E}\langle\boldsymbol{w}^s - \boldsymbol{w}^{s+1}, \frac{1}{b_s}\sum_{i=1}^{b_s} \boldsymbol{w}_i^s - \boldsymbol{w}^s\rangle + \mathbb{E}\|\boldsymbol{w}^{s+1} - \boldsymbol{w}^s\|^2\right]$$

$$\overset{\text{②}}{\leq} \mathbb{E}[F(\boldsymbol{w}^s)] + \frac{\lambda}{2}\left[1 - \frac{2}{\lambda\eta_s}\right]\mathbb{E}\|\boldsymbol{w}^{s+1} - \boldsymbol{w}^s\|^2$$

where ① holds since we sample the $b_s$ tasks uniformly from the observed $n$ tasks; ② uses the updating equation $\boldsymbol{w}^{s+1} = \boldsymbol{w}^s - \eta_s\lambda(\boldsymbol{w}^s - \frac{1}{b_s}\sum_{i=1}^{b_s}\boldsymbol{w}_{T_i}^s)$. Therefore, we have

$$\mathbb{E}\left[F(\boldsymbol{w}^{s+1}) - F(\boldsymbol{w}^s)\right] \leq \frac{\lambda}{2}\left[1 - \frac{2}{\lambda\eta_s}\right]\mathbb{E}\|\boldsymbol{w}^{s+1} - \boldsymbol{w}^s\|^2 < 0.$$

Then we sum up the above inequality and can further establish

$$\mathbb{E}\left[F(\boldsymbol{w}^0) - F(\boldsymbol{w}^{s+1})\right] \geq \frac{\lambda}{2}\left[\frac{2}{\lambda\eta_s} - 1\right]\sum_{i=0}^s \mathbb{E}\|\boldsymbol{w}^{s+1} - \boldsymbol{w}^s\|^2.$$

As $\frac{2}{\lambda \eta_s} - 1 > 0$ and $\inf_{\boldsymbol{w}} F(\boldsymbol{w}) > -\infty$, this implies $\lim_{s \to +\infty} \mathbb{E}[\|\boldsymbol{w}^{s+1} - \boldsymbol{w}^s\|] = 0$. That is, there exists a point $\boldsymbol{w}^* = \lim_{s \to +\infty} \mathbb{E}[\boldsymbol{w}^s]$. Therefore, according to the updating rule, we have $0 = \lim_{s \to +\infty} \mathbb{E}[\boldsymbol{w}^{s+1} - \boldsymbol{w}^s + \eta_s \lambda (\boldsymbol{w}^s - \frac{1}{b_s} \sum_{i=1}^{b_s} \boldsymbol{w}_{T_i}^s)] = \lim_{s \to +\infty} \mathbb{E}[\boldsymbol{w}^s - \frac{1}{b_s} \sum_{i=1}^{b_s} \boldsymbol{w}_{T_i}^s]$, implying $\nabla_{\boldsymbol{w}^*} F(\boldsymbol{w}^*) = \lim_{s \to +\infty} \nabla_{\boldsymbol{w}^s} F(\boldsymbol{w}^s) = \lim_{s \to +\infty} \mathbb{E}[\lambda (\boldsymbol{w}^s - \frac{1}{n} \sum_{i=1}^n \boldsymbol{w}_{T_i}^s)] = \lambda(\boldsymbol{w}^* - \frac{1}{n} \sum_{i=1}^n \boldsymbol{w}_{T_i}^*)] = 0$. This indicates that the sequence $\{\boldsymbol{w}^s\}$ will converge to a Karush–Kuhn–Tucker point. The proof is completed. $\qquad\square$

## C  Proof of The Results in Sec. 3.3

### C.1  Auxiliary Lemmas

In this section, we introduce auxiliary lemmas which will be used for proving the results in Sec. 3.3.

**Lemma 5.** *Assume that $\ell(f(\boldsymbol{w}_T, \boldsymbol{x}), \boldsymbol{y})$ is $L$-smooth in $\boldsymbol{w}_T$. If $\lambda > L$, then it holds for any $\boldsymbol{w}$ that*

$$\mathcal{L}_{D_T}(\boldsymbol{w}_T^*) - \mathcal{L}_{D_T}(\boldsymbol{w}) \le \frac{\lambda}{2} \|\boldsymbol{w}^* - \boldsymbol{w}\|^2 - \frac{\lambda - L}{2} \|\boldsymbol{w}_T^* - \boldsymbol{w}\|^2 - \frac{\lambda}{2} \|\boldsymbol{w}^* - \boldsymbol{w}_T^*\|^2. \tag{5}$$

*Moreover, assume that $\ell(f(\boldsymbol{w}_T, \boldsymbol{x}), \boldsymbol{y})$ is also convex in $\boldsymbol{w}_T$. Then for any $\boldsymbol{w}$ we have*

$$\mathcal{L}_{D_T}(\boldsymbol{w}_T^*) - \mathcal{L}_{D_T}(\boldsymbol{w}) \le \frac{\lambda}{2} \|\boldsymbol{w}^* - \boldsymbol{w}\|^2 - \frac{\lambda}{2} \|\boldsymbol{w}_T^* - \boldsymbol{w}\|^2 - \frac{\lambda}{2} \|\boldsymbol{w}_T^* - \boldsymbol{w}^*\|^2. \tag{6}$$

*Proof.* Let $\psi_{D_T}(\boldsymbol{w}_T) = \mathcal{L}_{D_T}(\boldsymbol{w}_T) + \frac{\lambda}{2} \|\boldsymbol{w}_T - \boldsymbol{w}\|_2^2$. Since $\ell(f(\boldsymbol{w}_T, \boldsymbol{x}), \boldsymbol{y})$ for all $T$ is $L$-smooth in $\boldsymbol{w}_T$ and $\boldsymbol{w}_T^*$ is optimal for $\psi(\boldsymbol{w}_T)$, it is straightforward to show that for any $\boldsymbol{w}$

$$\psi_{D_T}(\boldsymbol{w}) \ge \psi_{D_T}(\boldsymbol{w}_T^*) + \frac{\lambda - L}{2} \|\boldsymbol{w} - \boldsymbol{w}_T^*\|^2,$$

which leads to

$$\mathcal{L}_{D_T}(\boldsymbol{w}) \ge \mathcal{L}_{D_T}(\boldsymbol{w}_T^*) - \frac{\lambda}{2} \|\boldsymbol{w}^* - \boldsymbol{w}\|^2 + \frac{\lambda - L}{2} \|\boldsymbol{w}_T^* - \boldsymbol{w}\|^2 + \frac{\lambda}{2} \|\boldsymbol{w}^* - \boldsymbol{w}_T^*\|^2.$$

Moreover, if $\ell(f(\boldsymbol{w}_T, \boldsymbol{x}), \boldsymbol{y})$ is convex in $\boldsymbol{w}_T$, then we have that $\psi(\boldsymbol{w}_T^*)$ is $\lambda$-strongly convex. Based on the optimality of $\boldsymbol{w}_T^*$ we obtain that for any $\boldsymbol{w}$

$$\psi_{D_T}(\boldsymbol{w}) \ge \psi_{D_T}(\boldsymbol{w}_T^*) + \frac{\lambda - L}{2} \|\boldsymbol{w} - \boldsymbol{w}_T^*\|^2,$$

which implies

$$\mathcal{L}_{D_T}(\boldsymbol{w}) \ge \mathcal{L}_{D_T}(\boldsymbol{w}_T^*) - \frac{\lambda}{2} \|\boldsymbol{w}^* - \boldsymbol{w}\|^2 + \frac{\lambda}{2} \|\boldsymbol{w}_T^* - \boldsymbol{w}\|^2 + \frac{\lambda}{2} \|\boldsymbol{w}^* - \boldsymbol{w}_T^*\|^2.$$

The proof is completed. $\qquad\square$

The following lemma is a generalization of the result in [3].

**Lemma 6.** *Assume that $\ell(f(\boldsymbol{w}, \boldsymbol{x}), \boldsymbol{y})$ is $G$-Lipschitz continuous and $L$-smooth with respect to $\boldsymbol{w}$. Given a learning task $T$, let $\mathcal{L}(\boldsymbol{w}_T) = \mathbb{E}_{(\boldsymbol{x}, \boldsymbol{y}) \sim T}[\ell(f(\boldsymbol{w}_T, \boldsymbol{x}), \boldsymbol{y})]$ and $\mathcal{L}_{D_T}(\boldsymbol{w}_T) = \frac{1}{K} \sum_{(\boldsymbol{x}, \boldsymbol{y}) \in D_T} \ell(f(\boldsymbol{w}_T, \boldsymbol{x}), \boldsymbol{y})$ respectively denote the expected and empirical losses on $D_T = \{(\boldsymbol{x}_i, \boldsymbol{y}_i)\}_{i=1}^K \sim T$. Consider the following empirical minimization problem:*

$$\boldsymbol{w}_T^* = \operatorname*{argmin}_{\boldsymbol{w}_T} \left\{ \psi_{D_T}(\boldsymbol{w}_T) = \left\{ \mathcal{L}_{D_T}(\boldsymbol{w}_T) + \frac{\lambda}{2} \|\boldsymbol{w}_T - \boldsymbol{w}^*\|^2 \right\} \right\}.$$

*Then the following bound holds for if $\lambda > L$:*

$$\left| \mathbb{E}_{D_T \sim T}\left[ \mathcal{L}(\boldsymbol{w}_T^*) - \mathcal{L}_{D_T}(\boldsymbol{w}_T^*) \right] \right| \le \frac{4G^2}{(\lambda - L)K}, \quad \left\| \mathbb{E}_{D_T \sim T}\left[ \nabla \mathcal{L}(\boldsymbol{w}_T^*) - \nabla \mathcal{L}_{D_T}(\boldsymbol{w}_T^*) \right] \right\| \le \frac{4GL}{(\lambda - L)K}.$$

*Moreover, assume that $\ell(f(\boldsymbol{w}, \boldsymbol{x}), \boldsymbol{y})$ is convex. Then the following bound holds for any $\lambda > 0$:*

$$\left| \mathbb{E}_{D_T \sim T}\left[ \mathcal{L}(\boldsymbol{w}_T^*) - \mathcal{L}_{D_T}(\boldsymbol{w}_T^*) \right] \right| \le \frac{4G^2}{\lambda K}, \quad \left\| \mathbb{E}_{D_T \sim T}\left[ \nabla \mathcal{L}(\boldsymbol{w}_T^*) - \nabla \mathcal{L}_{D_T}(\boldsymbol{w}_T^*) \right] \right\| \le \frac{4GL}{\lambda K}.$$

*Proof.* The result can be proved by stability argument. For brevity, let $r(\boldsymbol{w}_T) = \frac{\lambda}{2}\|\boldsymbol{w}_T - \boldsymbol{w}^*\|^2$ is a $\lambda$-strongly convex regularization function. Let us consider $D_T^{(i)}$ which is identical to $D_T$ except that one of the $(\boldsymbol{x}_i, \boldsymbol{y}_i)$ is replaced by another random sample $(\boldsymbol{x}_i', \boldsymbol{y}_i')$. We then denote

$$\boldsymbol{w}_{T,i}^* = \operatorname*{argmin}_{\boldsymbol{w}_T} \left\{ \psi_{D_T^{(i)}}(\boldsymbol{w}_T) := \frac{1}{K}\left(\sum_{j\neq i} \ell(f(\boldsymbol{w}_T, \boldsymbol{x}_j), \boldsymbol{y}_j) + \ell(f(\boldsymbol{w}_T, \boldsymbol{x}_i'), \boldsymbol{y}_i')\right) + r(\boldsymbol{w}_T) \right\}.$$

Then we can show that

$$\psi_{D_T}(\boldsymbol{w}_{T,i}^*) - \psi_{D_T}(\boldsymbol{w}_T^*)$$

$$= \frac{1}{K}\sum_{j\neq i}\left(\ell(f(\boldsymbol{w}_{T,i}^*, \boldsymbol{x}_j), \boldsymbol{y}_j) - \ell(f(\boldsymbol{w}_T^*, \boldsymbol{x}_j), \boldsymbol{y}_j)\right) + \frac{1}{K}\left(\ell(f(\boldsymbol{w}_{T,i}^*, \boldsymbol{x}_i), \boldsymbol{y}_i) - \ell(f(\boldsymbol{w}_T^*, \boldsymbol{x}_i), \boldsymbol{y}_i)\right)$$

$$+ r(\boldsymbol{w}_{T,i}^*) - r(\boldsymbol{w}_T^*)$$

$$= \psi_{D_T^{(i)}}(\boldsymbol{w}_{T,i}^*) - \psi_{D_T^{(i)}}(\boldsymbol{w}_T^*) + \frac{1}{K}\left(\ell(f(\boldsymbol{w}_{T,i}^*, \boldsymbol{x}_i), \boldsymbol{y}_i) - \ell(f(\boldsymbol{w}_T^*, \boldsymbol{x}_i), \boldsymbol{y}_i)\right)$$

$$- \frac{1}{K}\left(\ell(f(\boldsymbol{w}_{T,i}^*, \boldsymbol{x}_i'), \boldsymbol{y}_i') - \ell(f(\boldsymbol{w}_T^*, \boldsymbol{x}_i'), \boldsymbol{y}_i')\right)$$

$$\overset{\text{①}}{\leq} \frac{1}{K}\left|\ell(f(\boldsymbol{w}_{T,i}^*, \boldsymbol{x}_i), \boldsymbol{y}_i) - \ell(f(\boldsymbol{w}_T^*, \boldsymbol{x}_i), \boldsymbol{y}_i)\right| + \frac{1}{K}\left|\ell(f(\boldsymbol{w}_{T,i}^*, \boldsymbol{x}_i'), \boldsymbol{y}_i') - \ell(f(\boldsymbol{w}_T^*, \boldsymbol{x}_i'), \boldsymbol{y}_i')\right|$$

$$\overset{\text{②}}{\leq} \frac{2G}{K}\|\boldsymbol{w}_T^* - \boldsymbol{w}_{T,i}^*\|,$$

where in ① we have used the optimality of $\boldsymbol{w}_{T,i}^*$ with respect to $\psi_{D_T^{(i)}}(\boldsymbol{w}_T)$, and in ② we use the Lipschitz continuity of the loss function $\ell$. Since $\ell$ is $L$-smooth and $\boldsymbol{w}_T^*$ is optimal for $\psi_{D_T}(\boldsymbol{w}_T)$, it is easily to verify that

$$\psi_{D_T}(\boldsymbol{w}_{T,i}^*) \geq \psi_{D_T}(\boldsymbol{w}_T^*) + \frac{\lambda - L}{2}\|\boldsymbol{w}_{T,i}^* - \boldsymbol{w}_T^*\|^2. \tag{7}$$

Provided that $\lambda > L$, by combing the preceding two inequalities we arrive at

$$\|\boldsymbol{w}_{T,i}^* - \boldsymbol{w}_T^*\| \leq \frac{4G}{(\lambda - L)K}.$$

It then follows consequently from the Lipschitz continuity of $\ell$ that for any sample $(\boldsymbol{x}, \boldsymbol{y}) \sim T$

$$|\ell(f(\boldsymbol{w}_{T,i}^*, \boldsymbol{x}), \boldsymbol{y}) - \ell(f(\boldsymbol{w}_T^*, \boldsymbol{x}), \boldsymbol{y})| \leq G\|\boldsymbol{w}_{T,i}^* - \boldsymbol{w}_T^*\| \leq \frac{4G^2}{(\lambda - L)K}. \tag{8}$$

Note that $D_T$ and $D_T^{(i)}$ are both i.i.d. samples of the task $T$. It follows that

$$\mathbb{E}_{D_T}[\mathcal{L}(\boldsymbol{w}_T^*)] = \mathbb{E}_{D_T^{(i)}}[\mathcal{L}(\boldsymbol{w}_{T,i}^*)] = \mathbb{E}_{D_T^{(i)} \cup \{(\boldsymbol{x}_i, \boldsymbol{y}_i)\}}[\ell(f(\boldsymbol{w}_{T,i}^*, \boldsymbol{x}_i), \boldsymbol{y}_i)].$$

Since the above holds for all $i = 1, ..., K$, we can show that

$$\mathbb{E}_{D_T}[\mathcal{L}(\boldsymbol{w}_T^*)] = \frac{1}{K}\sum_{i=1}^{K}\mathbb{E}_{D_T^{(i)} \cup \{(\boldsymbol{x}_i, \boldsymbol{y}_i)\}}[\ell(f(\boldsymbol{w}_{T,i}^*, \boldsymbol{x}_i), \boldsymbol{y}_i)] = \frac{1}{K}\sum_{i=1}^{K}\mathbb{E}_{D_T \cup \{(\boldsymbol{x}_i', \boldsymbol{y}_i')\}}[\ell(f(\boldsymbol{w}_{T,i}^*, \boldsymbol{x}_i), \boldsymbol{y}_i)].$$

Concerning the empirical case, we can see that

$$\mathbb{E}_{D_T}[\mathcal{L}_{D_T}(\boldsymbol{w}_T^*)] = \frac{1}{K}\sum_{i=1}^{K}\mathbb{E}_{D_T}[\ell(f(\boldsymbol{w}_T^*, \boldsymbol{x}_i), \boldsymbol{y}_i)] = \frac{1}{K}\sum_{i=1}^{K}\mathbb{E}_{D_T \cup \{(\boldsymbol{x}_i', \boldsymbol{y}_i')\}}[\ell(f(\boldsymbol{w}_T^*, \boldsymbol{x}_i), \boldsymbol{y}_i)].$$

By combining the above two inequalities we get

$$\left|\mathbb{E}_{D_T}[\mathcal{L}(\boldsymbol{w}_T^*) - \mathcal{L}_{D_T}(\boldsymbol{w}_{T,i}^*)]\right| = \left|\frac{1}{K}\sum_{i=1}^{K}\mathbb{E}_{D_T \cup \{(\boldsymbol{x}_i', \boldsymbol{y}_i')\}}[\ell(f(\boldsymbol{w}_T^*, \boldsymbol{x}_i), \boldsymbol{y}_i) - \ell(f(\boldsymbol{w}_{T,i}^*, \boldsymbol{x}_i), \boldsymbol{y}_i)]\right|$$

$$\leq \frac{1}{K}\sum_{i=1}^{K}\mathbb{E}_{D_T \cup \{(\boldsymbol{x}_i', \boldsymbol{y}_i')\}}\left[|\ell(f(\boldsymbol{w}_T^*, \boldsymbol{x}_i), \boldsymbol{y}_i) - \ell(f(\boldsymbol{w}_{T,i}^*, \boldsymbol{x}_i), \boldsymbol{y}_i)|\right]$$

$$\leq \frac{4G^2}{(\lambda - L)K},$$

where in the last inequality we have used (8). This proves the objective function inequality in the first part of the lemma. To prove the gradient norm inequality, we note from the smoothness assumption that

$$\|\nabla \ell(f(\boldsymbol{w}_T^*, \boldsymbol{x}), \boldsymbol{y}) - \nabla \ell(f(\boldsymbol{w}_{T,i}^*, \boldsymbol{x}), \boldsymbol{y})\| \le L\|\boldsymbol{w}_T^* - \boldsymbol{w}_{T,i}^*\| \le \frac{4GL}{(\lambda - L)K}. \tag{9}$$

The rest of the argument mimics that for the objective value case. Here we provide the details for the sake of completeness. Again, note that $D_T$ and $D_T^{(i)}$ are both i.i.d. samples of the task distribution $T$. It follows that

$$\mathbb{E}_{D_T}\left[\nabla \mathcal{L}(\boldsymbol{w}_T^*)\right] = \mathbb{E}_{D_T^{(i)}}\left[\nabla \mathcal{L}(\boldsymbol{w}_{T,i}^*)\right] = \mathbb{E}_{D_T^{(i)} \cup \{(\boldsymbol{x}_i, \boldsymbol{y}_i)\}}\left[\nabla \ell(f(\boldsymbol{w}_{T,i}^*, \boldsymbol{x}_i), \boldsymbol{y}_i)\right].$$

Since the above holds for all $i = 1, ..., m$, we can show that

$$\mathbb{E}_{D_T}\left[\nabla \mathcal{L}(\boldsymbol{w}_T^*)\right] = \frac{1}{K}\sum_{i=1}^{K}\mathbb{E}_{D_T^{(i)} \cup \{(\boldsymbol{x}_i, \boldsymbol{y}_i)\}}\left[\nabla \ell(f(\boldsymbol{w}_{T,i}^*, \boldsymbol{x}_i), \boldsymbol{y}_i)\right]$$

$$= \frac{1}{K}\sum_{i=1}^{K}\mathbb{E}_{D_T \cup \{(\boldsymbol{x}_i', \boldsymbol{y}_i')\}}\left[\nabla \ell(f(\boldsymbol{w}_{T,i}^*, \boldsymbol{x}_i), \boldsymbol{y}_i)\right].$$

Concerning the empirical version, we can see that

$$\mathbb{E}_{D_T}\left[\nabla \mathcal{L}_{D_T}(\boldsymbol{w}_T^*)\right] = \frac{1}{K}\sum_{i=1}^{K}\mathbb{E}_{D_T}\left[\nabla \ell(f(\boldsymbol{w}_T^*, \boldsymbol{x}_i), \boldsymbol{y}_i)\right] = \frac{1}{K}\sum_{i=1}^{K}\mathbb{E}_{D_T \cup \{(\boldsymbol{x}_i', \boldsymbol{y}_i')\}}\left[\nabla \ell(f(\boldsymbol{w}_T^*, \boldsymbol{x}_i), \boldsymbol{y}_i)\right].$$

By combining the above two inequalities we get

$$\left\|\mathbb{E}_{D_T}\left[\nabla \mathcal{L}(\boldsymbol{w}_T^*) - \nabla \mathcal{L}_{D_T}(\boldsymbol{w}_{T,i}^*)\right]\right\|$$

$$= \left\|\frac{1}{K}\sum_{i=1}^{K}\mathbb{E}_{D_T \cup \{(\boldsymbol{x}_i', \boldsymbol{y}_i')\}}\left[\nabla \ell(f(\boldsymbol{w}_T^*, \boldsymbol{x}_i), \boldsymbol{y}_i) - \nabla \ell(f(\boldsymbol{w}_{T,i}^*, \boldsymbol{x}_i), \boldsymbol{y}_i)\right]\right\|$$

$$\le \frac{1}{K}\sum_{i=1}^{K}\mathbb{E}_{D_T \cup \{(\boldsymbol{x}_i', \boldsymbol{y}_i')\}}\left[\left\|\nabla \ell(f(\boldsymbol{w}_T^*, \boldsymbol{x}_i), \boldsymbol{y}_i) - \nabla \ell(f(\boldsymbol{w}_{T,i}^*, \boldsymbol{x}_i), \boldsymbol{y}_i)\right\|\right]$$

$$\le \frac{4GL}{(\lambda - L)K},$$

where in the last inequality we have used (9).

To prove the second part, we can just apply the almost identical stability argument except that the inequality (7) can now be replaced by a stronger version due to the convexity of $\ell$:

$$\psi_{D_T}(\boldsymbol{w}_{T,i}^*) \ge \psi_{D_T}(\boldsymbol{w}_T^*) + \frac{\lambda}{2}\|\boldsymbol{w}_{T,i}^* - \boldsymbol{w}_T^*\|^2.$$

The proof is concluded. $\qquad\square$

## C.2 Proof of Theorem 2

*Proof.* Consider a fixed task $T \sim \mathcal{T}$ and its associated random sample $D_T \sim T$ of size $K$. We denote $\mathcal{L}_{D_T}(\boldsymbol{w}) = \frac{1}{K}\sum_{(\boldsymbol{x}, \boldsymbol{y}) \in D_T} \ell(f(\boldsymbol{w}_T, \boldsymbol{x}), \boldsymbol{y})$. From Lemma 6 we know that

$$\left|\mathbb{E}_{D_T \sim T}\left[\mathcal{L}(\boldsymbol{w}_T^*) - \mathcal{L}_{D_T}(\boldsymbol{w}_T^*)\right]\right| \le \frac{4G^2}{\lambda K}. \tag{10}$$

From Lemma 5, for any $\boldsymbol{w}$ we have

$$\mathcal{L}_{D_T}(\boldsymbol{w}_T^*) - \mathcal{L}_{D_T}(\boldsymbol{w}) \le \frac{\lambda}{2}\|\boldsymbol{w}^* - \boldsymbol{w}\|^2 - \frac{\lambda}{2}\|\boldsymbol{w}_T^* - \boldsymbol{w}\|^2 - \frac{\lambda}{2}\|\boldsymbol{w}^* - \boldsymbol{w}_T^*\|^2. \tag{11}$$

By taking expectation over the random sample set $D_T$ at $\boldsymbol{w} = \boldsymbol{w}_{T,E}^*$ we obtain

$$
\begin{aligned}
\mathbb{E}_{D_T}[\mathcal{L}_{D_T}(\boldsymbol{w}_T^*) - \mathcal{L}_{D_T}(\boldsymbol{w}_{T,E}^*)] \leq & \mathbb{E}_{D_T}\left[\frac{\lambda}{2}\|\boldsymbol{w}^* - \boldsymbol{w}_{T,E}^*\|^2 - \frac{\lambda}{2}\|\boldsymbol{w}_T^* - \boldsymbol{w}_{T,E}^*\|^2 - \frac{\lambda}{2}\|\boldsymbol{w}^* - \boldsymbol{w}_T^*\|^2\right] \\
\leq & \frac{\lambda}{2}\mathbb{E}_{D_T}\left[\|\boldsymbol{w}^* - \boldsymbol{w}_{T,E}^*\|^2\right].
\end{aligned}
\tag{12}
$$

Then we can show the following

$$
\begin{aligned}
\mathbb{E}_{D_T}\left[\mathcal{L}(\boldsymbol{w}_T^*) - \mathcal{L}(\boldsymbol{w}_{T,E}^*)\right] = & \mathbb{E}_{D_T}\left[\mathcal{L}(\boldsymbol{w}_T^*) - \mathcal{L}_{D_T}(\boldsymbol{w}_{T,E}^*)\right] + \mathbb{E}_{D_T}\left[\mathcal{L}_{D_T}(\boldsymbol{w}_T^*) - \mathcal{L}(\boldsymbol{w}_{T,E}^*)\right] \\
\leq & \left|\mathbb{E}_{D_T}\left[\mathcal{L}(\boldsymbol{w}_T^*) - \mathcal{L}_{D_T}(\boldsymbol{w}_{T,E}^*)\right]\right| + \mathbb{E}_{D_T}\left[\mathcal{L}_{D_T}(\boldsymbol{w}_T^*) - \mathcal{L}(\boldsymbol{w}_{T,E}^*)\right] \\
\overset{\text{①}}{\leq} & \frac{4G^2}{\lambda K} + \frac{\lambda}{2}\mathbb{E}_{D_T}\left[\|\boldsymbol{w}^* - \boldsymbol{w}_{T,E}^*\|^2\right],
\end{aligned}
$$

where in the last inequality we have used Eqn. (10) and the above inequality (12). Now we can take expectation of both sides of the above over $T \sim \mathcal{T}$ to obtain

$$
\mathbb{E}_{T\sim\mathcal{T}}\mathbb{E}_{D_T}\left[\mathcal{L}(\boldsymbol{w}_T^*) - \mathcal{L}(\boldsymbol{w}_{T,E}^*)\right] \leq \frac{4G^2}{\lambda K} + \frac{\lambda}{2}\mathbb{E}_{T\sim\mathcal{T}}\left[\|\boldsymbol{w}^* - \boldsymbol{w}_{T,E}^*\|^2\right].
$$

This proves the results in the theorem. $\qquad\square$

### C.3    Proof of Theorem 3

*Proof.* Consider a fixed task $T \sim \mathcal{T}$ and its associated random sample $D_T \sim T$ of size $K$. From the smoothness of $\ell(f(\boldsymbol{w}, \boldsymbol{x}), \boldsymbol{y})$ we can derive that

$$
\begin{aligned}
\mathcal{L}_{D_T}(\boldsymbol{w}^*) \geq & \mathcal{L}_{D_T}(\boldsymbol{w}_T^*) + \langle\nabla\mathcal{L}_{D_T}(\boldsymbol{w}_T^*), \boldsymbol{w}^* - \boldsymbol{w}_T^*\rangle - \frac{L}{2}\|\boldsymbol{w}^* - \boldsymbol{w}_T^*\|^2 \\
= & \mathcal{L}_{D_T}(\boldsymbol{w}_T^*) + \frac{1}{\lambda}\|\nabla\mathcal{L}_{D_T}(\boldsymbol{w}_T^*)\|^2 - \frac{L}{2\lambda^2}\|\nabla\mathcal{L}_{D_T}(\boldsymbol{w}_T^*)\|^2 \\
\geq & \mathcal{L}_{D_T}(\boldsymbol{w}_T^*) + \frac{1}{\lambda}\left[1 - \frac{L}{2\lambda}\right]\|\nabla\mathcal{L}_{D_T}(\boldsymbol{w}_T^*)\|^2,
\end{aligned}
\tag{13}
$$

where we have used the first-order optimality condition $\nabla\mathcal{L}_{D_T}(\boldsymbol{w}_T^*) + \lambda(\boldsymbol{w}_T^* - \boldsymbol{w}^*) = 0$ and $\lambda > L$. Then we can show the following

$$
\begin{aligned}
& \left\|\mathbb{E}_{D_T}\left[\nabla\mathcal{L}(\boldsymbol{w}_T^*)\right]\right\|^2 \\
= & \left\|\mathbb{E}_{D_T}\left[\nabla\mathcal{L}(\boldsymbol{w}_T^*) - \nabla\mathcal{L}_{D_T}(\boldsymbol{w}_T^*)\right] + \mathbb{E}_{D_T}\left[\nabla\mathcal{L}_{D_T}(\boldsymbol{w}_T^*)\right]\right\|^2 \\
\leq & 2\left\|\mathbb{E}_{D_T}\left[\nabla\mathcal{L}(\boldsymbol{w}_T^*) - \nabla\mathcal{L}_{D_T}(\boldsymbol{w}_T^*)\right]\right\|^2 + 2\left\|\mathbb{E}_{D_T}\left[\nabla\mathcal{L}_{D_T}(\boldsymbol{w}_T^*)\right]\right\|^2 \\
\leq & 2\left\|\mathbb{E}_{D_T}\left[\nabla\mathcal{L}(\boldsymbol{w}_T^*) - \nabla\mathcal{L}_{D_T}(\boldsymbol{w}_T^*)\right]\right\|^2 + 2\mathbb{E}_{D_T}\left[\left\|\nabla\mathcal{L}_{D_T}(\boldsymbol{w}_T^*)\right\|^2\right] \\
\overset{\text{①}}{\leq} & 2\left\|\mathbb{E}_{D_T}\left[\nabla\mathcal{L}(\boldsymbol{w}_T^*) - \nabla\mathcal{L}_{D_T}(\boldsymbol{w}_T^*)\right]\right\|^2 + \frac{2}{\beta}\mathbb{E}_{D_T}\left[\mathcal{L}_{D_T}(\boldsymbol{w}^*) - \mathcal{L}_{D_T}(\boldsymbol{w}_T^*)\right] \\
= & 2\left\|\mathbb{E}_{D_T}\left[\nabla\mathcal{L}(\boldsymbol{w}_T^*) - \nabla\mathcal{L}_{D_T}(\boldsymbol{w}_T^*)\right]\right\|^2 + \frac{2}{\beta}\mathbb{E}_{D_T}\left[\mathcal{L}_{D_T}(\boldsymbol{w}^*) - \mathcal{L}(\boldsymbol{w}_T^*)\right] + \frac{2}{\beta}\mathbb{E}_{D_T}\left[\mathcal{L}(\boldsymbol{w}_T^*) - \mathcal{L}_{D_T}(\boldsymbol{w}_T^*)\right] \\
\overset{\text{②}}{\leq} & \frac{32G^2L^2}{(\lambda - L)^2K^2} + \frac{8G^2}{(\lambda - L)\beta K} + \frac{2}{\beta}\mathbb{E}_{D_T}\left[\mathcal{L}_{D_T}(\boldsymbol{w}^*) - \mathcal{L}(\boldsymbol{w}_T^*)\right] \\
= & \frac{32G^2L^2}{(\lambda - L)^2K^2} + \frac{8G^2}{(\lambda - L)\beta K} + \frac{2}{\beta}\left[\mathcal{L}(\boldsymbol{w}^*) - \mathcal{L}(\boldsymbol{w}_T^*)\right] \\
\overset{\text{③}}{\leq} & \frac{32G^2L^2}{(\lambda - L)^2K^2} + \frac{8G^2}{(\lambda - L)\beta K} + \frac{2}{\beta}\left[\mathcal{L}(\boldsymbol{w}^*) - \mathcal{L}(\boldsymbol{w}_{T,E}^*)\right],
\end{aligned}
$$

where in ① we used inequality (13) and let $\beta = \frac{1}{\lambda}\left[1 - \frac{L}{2\lambda}\right]$, in ② we used Lemma 6:

$$
\left|\mathbb{E}_{D_T\sim T}\left[\mathcal{L}(\boldsymbol{w}_T^*) - \mathcal{L}_{D_T}(\boldsymbol{w}_T^*)\right]\right| \leq \frac{4G^2}{(\lambda - L)K}, \quad \left\|\mathbb{E}_{D_T\sim T}\left[\nabla\mathcal{L}(\boldsymbol{w}_T^*) - \nabla\mathcal{L}_{D_T}(\boldsymbol{w}_T^*)\right]\right\| \leq \frac{4GL}{(\lambda - L)K}.
$$

Figure 2: Effects of $\lambda$ to Meta-MinibatchProx on miniImageNet.

In ③, we use the fact that $\boldsymbol{w}_{T,E}^*$ is the optimum to the expected risk $\mathcal{L}(\boldsymbol{w})$. Now we can take expectation of both sides of the above over $T \sim \mathcal{T}$ to obtain

$$\mathbb{E}_{T\sim\mathcal{T}}\left[\left\|\mathbb{E}_{D_T\sim T}\left[\nabla\mathcal{L}(\boldsymbol{w}_T^*)\right]\right\|^2\right] \leq \frac{32G^2L^2}{(\lambda-L)^2K^2} + \frac{8G^2}{(\lambda-L)\beta K} + \frac{2}{\beta}\mathbb{E}_{T\sim\mathcal{T}}\left[\mathcal{L}(\boldsymbol{w}^*) - \mathcal{L}(\boldsymbol{w}_{T,E}^*)\right].$$

This completes the proof. $\qquad\qquad\square$

### C.4  Proof of Theorem 5

*Proof.* The proof of non-convex loss is very similar to the proof of convex case in Sec. C.2. For non-convex setting, we firstly replace the results in Eqn. (10) in Sec. C.2 by the first result in Lemma 6:

$$\left|\mathbb{E}_{D_T\sim T}\left[\mathcal{L}(\boldsymbol{w}_T^*) - \mathcal{L}_{D_T}(\boldsymbol{w}_T^*)\right]\right| \leq \frac{4G^2}{(\lambda-L)K}.$$

Then, we replace the results in Eqn. (11) in Sec. C.2 by the first result in Lemma 5 that for any $\boldsymbol{w}$ we have

$$\mathcal{L}_{D_T}(\boldsymbol{w}_T^*) - \mathcal{L}_{D_T}(\boldsymbol{w}) \leq \frac{\lambda}{2}\|\boldsymbol{w}^* - \boldsymbol{w}\|^2 - \frac{\lambda-L}{2}\|\boldsymbol{w}_T^* - \boldsymbol{w}\|^2 - \frac{\lambda}{2}\|\boldsymbol{w}^* - \boldsymbol{w}_T^*\|^2.$$

Then the following proof can be derived based on almost identical argument in Sec. C.2 under the assumption $\lambda > L$. In this way, we can obtain the desired result:

$$\mathbb{E}_{T\sim\mathcal{T}}\mathbb{E}_{D_T}\left[\mathcal{L}(\boldsymbol{w}_T^*) - \mathcal{L}(\boldsymbol{w}_{T,E}^*)\right] \leq \frac{4G^2}{(\lambda-L)K} + \frac{\lambda}{2}\mathbb{E}_{T\sim\mathcal{T}}\left[\|\boldsymbol{w}^* - \boldsymbol{w}_{T,E}^*\|^2\right].$$

The proof is completed. $\qquad\qquad\square$

## D  More Experimental Results

### D.1  Robust Evaluation Experiments on Classification Tasks

We also report the effects of $\lambda$ to the testing performance of our method in Fig. 2. When the value of $\lambda$ ranges from $10^{-1}$ to $10^{1.7}$, the performance of our method on miniImageNet are relatively stable. This well demonstrates the robustness of Meta-MinibatchProx to the choice of $\lambda$.

### D.2  More Experimental Results on Regression Tasks

Here we provide more experimental results for regression task. All the experimental setting is the same in the manuscript for regression task. By observing Fig. 3, we can find that MAML outperforms than its first-order variants, namely FOMAML and Reptile. Moveover, one can observe that our proposed Meta-MinibatchProx also outperforms all other approaches including MAML. These results are consistent with the visual results and numerical results in the manuscript. All results shows the advantages of our proposed Meta-MinibatchProx approach.

Figure 3: The illustration of the compared meta learning methods on the few-shot regression problem.