[Reviews · NeurIPS 2019]

Reviewer 1



Originality: Admittedly (by the authors), the proposed algorithm is a fairly straightforward extension of previous ideas. However, the authors are up-front about this, and their manuscript contains considerably additional analysis and empirical work, which more than exceeds any sort of originality bar in this reviewer's opinion. Quality: The work is of high quality, with extensive theoretical analysis of the convergence of their algorithm. Clarity: The paper is quite clear. The appendix discussion of the proofs could use slightly more detail (particularly for outsides)--even a few sentences of guiding intuition could be useful (i.e., regarding Karush-Kuhn-Tucker points, for the uninitiated) Significance: This method clearly outperforms other methods, and is conceptually simpler and easier to compute. Conditioned on the authors actually providing tensorflow code for the result, this looks like a powerful, general technique for few-shot metalearning.

Reviewer 2



UPDATE: I'd like to thank the authors for their detailed response. In light of this response I have increased my score to a 7. Originality This paper presents Meta-MinibatchProx, an algorithm for model and algorithm agnostic meta learning that, unlike MAML and friends, comes with theoretical guarantees of convergence. To the best of my knowledge it is the first such algorithm to offer any convergence guarantees, and also has the potential to scale to very large problems. Quality Something I would like to have seen addressed explicitly in this paper is the distinction between what I will call the "finite" and "infinite" versions of MMP. The distinction here is related to the comment "Usually we are only provided with n observed tasks..." on line 135. In particular: 1. The number of tasks n is fairly small (say 10s) and we can permanently materialize the per task parameters w_T for each task (the "finite" case). 2. The number of tasks n is infinite (as in the sin wave experiments) or effectively so (as in the few shot image net experiments) where a single task is unlikely to be seen more than once, so permanent materialization of the task specific parameters is not useful (the "infinite" case). The paper focuses exclusively on the second setting (which is the typical framing for meta learning), but it seems to me like one of the big advantages of MMP is its ability to deal well with the finite setting as well. In the finite case I think the MMP framework offers a starting point for thinking about how to deal with issues like: 1. Task spaces where some tasks have vastly more data than others 2. Hierarchically structured task spaces (for example, the structured labels in tieredImageNet could be used to construct hierarchical priors). 3. Task spaces with non-uniform input or output structures, where task specific models share only some of their parameters (perhaps also with a graph structure over 4. dependencies between model parts) 5. And so on… This paper presents itself as a better MAML than MAML, which is justified through convergence theory and empirical results. It would be vastly more compelling if it also demonstrated the capability to deal with problems that are hard or impossible to express with MAML. I feel that not doing so is a quite a missed opportunity. Clarity I found the paper quite clear and well presented. There is necessarily a lot of notation in a paper of this sort, but I found it well presented and reasonably easy to follow. Significance As mentioned in the quality section, I think the biggest weakness of this paper is a missed opportunity to present an algorithm that can do more than the existing meta learning formulations. It is one thing to have a slightly better meta learning algorithm than the baselines, it is quite another to have that in a framework that is also general enough to address a wider class of problems. Typos 139: "inra-meta-task" 271: The text says 15 steps of SGD, but the legend in Figure 1(a) says 32 steps.

Reviewer 3



UPDATE: After reading the author feedback I would like to keep my score and stick to the recommendation of introducing a more intuitive, graphical explanation of the difference between MMP and MAML, similarly to how my question has been answered. Novelty: The approach seems to be quiet novel, as well as the convergence results. I am not an expert in this, but I think it should not be difficult to provide similar converge guarantees e.g. for the standard MAML, and their absence is mostly due to lack of interest from the community. However, it is important that finally such results are available. Quality: The proposed Meta-MinimatchProx is a sound framework, analysed well both from theoretical and experimental perspectives. I have only a few questions / suggestions to the authors. I would appreciate more discussion of the parameter lambda and how one should set it. It must have something to do with the (meta-)generalisation performance as it is the strength of the prior in some sense, and if it’s the case it would be interesting to see a graph of the test performance, e.g. on ImageNet, depending on lambda. Clarity: The paper is written well and generally easy to follow. I have only a few comments: 1. The distinction between MAML and Meta-MinimatchProx updates discussed on lines 132-151 could be made clearer and perhaps supplied with a graphical illustration. 2. It looks like function F(w) appears first in eq. 3 and is defined implicitly. Significance: The paper makes an interesting contribution to a very important area of meta-learning and, I think, will be recognised by the community.

[Author Response · NeurIPS 2019]

**General Response.** We thank all the reviewers for their insightful and encouraging comments. Below we provide our point-by-point response to the main concerns raised by the reviewers.

**To Reviewer #1.** Per your suggestion, we will update the appendix by adding more explanations about the proof ideas.

**To Reviewer #2.**

1) Since in many real applications, e.g. image classification, the task number $n$ is finite though could be large, w.l.o.g. we choose to focus on the finite setting (**FS**). But all the convergence and generalization guarantees in this work can be extended to the infinite setting (**IFS**) which will be emphasized in revision. We briefly introduce the idea of extension from FS to IFS. **For convergence**, the technical Lemmas $1 \sim 4$ hold for both settings as they do not involve FS and IFS. Let $\phi_{D_{T_i}}(\boldsymbol{w}) = \min_{\boldsymbol{w}_{T_i}} \mathcal{L}_{D_{T_i}}(\boldsymbol{w}_{T_i}) + \frac{\lambda}{2}\|\boldsymbol{w}_{T_i} - \boldsymbol{w}\|_2^2$ and $\boldsymbol{w}_{T_i}^* = \operatorname{argmin}_{\boldsymbol{w}_{T_i}} \mathcal{L}_{D_{T_i}}(\boldsymbol{w}_{T_i}) + \frac{\lambda}{2}\|\boldsymbol{w}_{T_i} - \boldsymbol{w}\|_2^2$. Extending Theorem 1 from FS to IFS only needs to extend (a) $\mathbb{E}[\frac{1}{b_s}\sum_{i=1}^{b_s} \phi_{D_{T_i}}(\boldsymbol{w})] = F(\boldsymbol{w})$ and (b) $\mathbb{E}[\frac{1}{b_s}\sum_{i=1}^{b_s} \nabla\phi_{D_{T_i}}(\boldsymbol{w})] = \nabla F(\boldsymbol{w})$ with $F(\boldsymbol{w}) = \frac{1}{n}\sum_{i=1}^{n} \phi_{D_{T_i}}(\boldsymbol{w})$ under FS respectively to (a) and (b) with $F(\boldsymbol{w}) = \mathbb{E}_{T \sim \mathcal{T}}\phi_{D_T}(\boldsymbol{w})$ for IFS. By sampling mini-batch $\{T_i\}$ as $T_i \sim \mathcal{T}$, (a) and (b) hold for IFS. As tasks $T_i$, e.g. in image classification, are usually from a uniform distribution $\mathcal{T}$, we can uniformly sample task $T_i$. The remaining proofs for IFS and FS are identical. Similarly, we can extend convergence results in Theorem 4 in Appendix from FS to IFS. **For generalization**, Theorems 2 and 3 still hold for IFS without needing any changes, as they provide generalization performance guarantee of empirical solution in any task $T \sim \mathcal{T}$.

When task number $n$ is fairly small, we agree that it is an interesting future work to explore the structure of task space, e.g. hierarchical structure. We expect that the approach developed in this paper will fuel this future investigation.

2) One advantage of MMP over MAML is that it can easily and flexibly consider the structures of solution space of tasks by designing proper $\|\boldsymbol{w}_{T_i} - \boldsymbol{w}\|_p^q$ so as to find better $\boldsymbol{w}_{T_i} = \operatorname{argmin}_{\boldsymbol{w}_{T_i}} \mathcal{L}_{D_{T_i}}(\boldsymbol{w}_{T_i}) + \frac{\lambda}{2}\|\boldsymbol{w}_{T_i} - \boldsymbol{w}\|_p^q$ and thus better prior $\boldsymbol{w}$. In contrast, MAML uses a fixed gradient descent update rule $\boldsymbol{w}_{T_i} = \boldsymbol{w} - \eta\nabla\mathcal{L}_{D_T}(\boldsymbol{w})$ according to its model $\mathcal{L}_{D_T}(\boldsymbol{w} - \eta\nabla\mathcal{L}_{D_T}(\boldsymbol{w}))$, hampering designing more flexible relation between $\boldsymbol{w}_{T_i}$ and $\boldsymbol{w}$. For instance, assume there are a few outlier tasks $\mathcal{O} = \{T_o\}$ whose optima $\boldsymbol{w}_o$ are far away from optima $\boldsymbol{w}_s$ of normal tasks $\mathcal{S} = \{T_s\}$. To handle this case, MMP can use the robust $\ell_{2,1}$ norm, i.e. $\frac{1}{n}\sum_{i=1}^{n}\|\boldsymbol{w}_{T_i} - \boldsymbol{w}\|_2$, to tolerate larger distances between $\boldsymbol{w}$ and some outlier optima $\boldsymbol{w}_{T_i}$, and the learned prior $\boldsymbol{w}$ is still close to optima $\boldsymbol{w}_s$ in $\mathcal{S}$ and only requires a few training data for adaptation to new normal tasks. In contrast, it is hard to tailor MAML to handle this case due to its fixed update rule. So being affected by outlier tasks, prior $\boldsymbol{w}$ departures away from $\boldsymbol{w}_s$ and needs more data for adaptation to new normal tasks. To verify this, let us consider an example where $5\%$ outlier images with zero pixels are added into each class in miniImageNet to form outlier tasks. As shown in Fig. 2, our experimental results justify that the outlier tasks can be well handled by MMP+$\ell_{21}$ to achieve robust meta-learning. We will update this into the revision.

3) We would like to clarify that the step number 15 mentioned in the text is used in the meta-training phase, while the step number 32 mentioned in Fig. 1 (of the submission) is used for fine-tuning (meta-test).

Fig. 2. Performance comparison on 1-shot 5-way outlier-corrupted data .    Fig. 3. Impact of $\lambda$ to classification accuracy on miniImageNet.

**To Reviewer #3.**

1) We report the impact of $\lambda$ on the testing performance of our method in Fig. 3. When the value of $\lambda$ ranges from $10^{-1}$ to $10^{1.7}$, the performance of our method are relatively stable, demonstrating its insensitivity to the choice of $\lambda$.

2) To highlight the difference between MAML and Meta-MinibatchProx (**MMP**), MAML aims to find an initialization $\boldsymbol{w}$ such that $\boldsymbol{w}_T^* = \boldsymbol{w} - \eta\nabla\mathcal{L}_{D_T}(\boldsymbol{w}) = \operatorname{argmin}_{\boldsymbol{w}_T}\langle\nabla\mathcal{L}_{D_T}(\boldsymbol{w}), \boldsymbol{w}_T - \boldsymbol{w}\rangle + \frac{1}{2\eta}\|\boldsymbol{w}_T - \boldsymbol{w}\|_2^2$ is close to the optimal hypothesis of task $T$. Differently, **MMP** is defined to find the task-specific optimal hypothesis by computing $\widetilde{\boldsymbol{w}}_T^* = \min_{\boldsymbol{w}_T} \mathcal{L}_{D_T}(\boldsymbol{w}_T) + \frac{\lambda}{2}\|\boldsymbol{w}_T - \boldsymbol{w}\|_2^2$. Essentially speaking, MAML approximates the loss $\mathcal{L}_{D_T}(\boldsymbol{w}_T)$ using its first-order taylor expansion for computing an approximate optimum $\boldsymbol{w}_T^*$; while MMP directly optimizes $\mathcal{L}_{D_T}(\boldsymbol{w}_T) = \langle\nabla\mathcal{L}_{D_T}(\boldsymbol{w}), \boldsymbol{w}_T - \boldsymbol{w}\rangle + \frac{1}{2}\langle\nabla^2\mathcal{L}_{D_T}(\boldsymbol{w})(\boldsymbol{w}_T - \boldsymbol{w}), (\boldsymbol{w}_T - \boldsymbol{w})\rangle + \frac{1}{6}\langle\nabla^3\mathcal{L}_{D_T}(\boldsymbol{w}), (\boldsymbol{w}_T - \boldsymbol{w})^{\otimes^3}\rangle + \cdots$. Therefore, **MMP** is able to make use of higher-order information of $\mathcal{L}_{D_T}$ beyond gradient to search optimal hypothesis around the prior $\boldsymbol{w}$, which could lead to better task-specific hypothesis and the prior hypothesis as well.

[Meta-Review · NeurIPS 2019]

All three reviewers agree that the paper is novel, well presented and shows convincing empirical results. The extensive theoretical analysis of the convergence of their algorithm makes the paper particularly interesting and different from prior related art. I agree with Reviewer 3, and encourage the authors to add a high level explanation of the difference between MMP and MAML, similar to the one provided in the author's response.